# TDP-43 upregulates lipid metabolism modulator ABHD2 to suppress apoptosis in hepatocellular carcinoma

Bo-wen Liu [1,2,3✉], Xiang-yun Wang[1,2], Jin-ling Cao[1,2], Lu-lu Chen[1], Yi-lei Wang[1], Bing-qian Zhao[1], Jia Zhou[1] & Zhi-fa Shen [1,3✉]

TAR DNA-Binding Protein 43 (TDP-43) has been well studied in neurodegenerative diseases, but its potential role in malignance is still unclear. Here, we demonstrate that TDP-43 contributes to the suppression of apoptosis by facilitating lipid metabolism in hepatocellular carcinoma (HCC). In HCC cells, TDP-43 is able to suppress apoptosis while deletion of it markedly induces apoptosis. RNA-sequencing identifies the lipid metabolism gene *abhydrolase domain containing 2 (ABHD2)* as the target gene of TDP-43. Tissue microarray analysis shows the positive correlation of TDP-43 and ABHD2 in HCC. Mechanistically, TDP-43 binds with the UG-rich sequence1 of *ABHD2* 3′UTR to enhance the mRNA stability of *ABHD2*, thereby upregulating ABHD2. Afterwards, TDP-43 promotes the production of free fatty acid and fatty acid oxidation-originated reactive oxygen species (ROS) in an ABHD2-dependent manner, so as to suppress apoptosis of HCC. Our findings provide insights into the mechanism of HCC progression and reveal TDP-43/ABHD2 as potential targets for the precise treatment of HCC.

---

[1] Henan Key Laboratory of Immunology and Targeted Drugs, School of Laboratory Medicine, Xinxiang Medical University, Xinxiang 453003, PR China. [2] These authors contributed equally: Bo-wen Liu, Xiang-yun Wang, Jin-ling Cao. [3] These authors jointly supervised this work: Bo-wen Liu, Zhi-fa Shen. ✉email: liubowen309@163.com; shenzhifa@wmu.edu.cn

Hepatocellular carcinoma (HCC), one of the common malignant tumors around the world[1,2], has seriously endangered human life and health for many years. The pathogenesis of HCC is complicated, it involves a variety of processes, such as chronic inflammation, liver fibrosis and oncogene dysregulation[2,3]. The dysregulation of oncogene expression always induces the disorder of the physiological state of cell, which results in excessive proliferation and suppressive apoptosis, accelerating the malignant progression of HCC[4]. Therefore, clarifying the mechanism of initiation and development is crucial for the targeted therapy in HCC.

Transactivation (TAR) DNA-binding protein (TARDBP), also known as TDP-43[5,6], was originally identified as a DNA-binding protein capable of regulating *human immunodeficiency virus 1* gene expression[7,8]. Later studies found that TDP-43, encoded by the *tardbp* gene, is a highly conserved DNA/RNA-binding protein. In the nucleus, TDP-43 can bind to RNA and play an important role in mRNA transcription, splicing, nuclear transport and stability[5,9]. In the cytoplasm, TDP-43 forms RNP ribonucleoprotein granules that regulate mRNA transport and promote the production of non-coding RNAs such as miRNA[10,11]. TDP-43 has been focused because of its vital role in neurodegenerative diseases, such as amyotrophic lateral sclerosis and Alzheimer's disease[12,13]. Recently, it has been discovered that TDP-43 is closely correlated with the progression of malignant tumors. Studies have shown that TDP-43 is dysregulated in breast cancer, lung cancer, melanoma and other malignant tumors[14–16]. In breast tumor, TDP-43 can interact with serine and arginine rich splicing factor 3 to modulate the selective splicing of downstream targets *PAPR3* and *NUMB* mRNA, thereby promoting tumor proliferation and metastasis[14]. In lung cancer cells, TDP-43 enhances the mRNA stability of *FasL* by interacting with *FasL* mRNA, thereby mediating apoptosis and finally inhibiting the progression of lung cancer[16].

Notably, a previous study shows that TDP-43 elevates the expression of the platelet isoform of phosphofructokinase, a rate-limiting enzyme of glycolysis, to promote cell growth in HCC[17]. This evidence indicates the potential role of TDP-43 in modulating cell metabolism in HCC. Abnormal metabolism is a common hallmark of cancer cells; especially glucose metabolism and lipid metabolism, cancer cells often reprogram these metabolic pathways to meet the energy needs of overgrowth and to resist apoptosis[18]. Liver is an important place for lipid metabolism, and plays a vital role in the absorption, synthesis, decomposition, and transport of lipids[19]. lipid metabolism disorder usually appears during the occurrence and development of HCC[20,21]. However, whether TDP-43 has an effect on lipid metabolism in the progression of HCC and the underlying mechanism are still unknown.

In the present study, we uncover that the expression of TDP-43 is negatively related to the overall survival and relapse-free survival of HCC. TDP-43 suppresses the cell apoptosis of HCC through the lipid metabolism-related gene *ABHD2*. Molecular mechanism investigation reveals that TDP-43 can bind to the 3′UTR of *ABHD2* mRNA and enhance the stability of *ABHD2* mRNA, thereby upregulating the expression of ABHD2. TDP-43 was able to promote the production of free fatty acid (FFA) and fatty acid oxidation-originated reactive oxygen species (ROS) in an ABHD2-dependent manner, thereby promoting proliferation and inhibiting apoptosis in HCC cells. Our study provides a potential target for the precise treatment of HCC.

## Results

### TDP-43 inhibits apoptosis and promotes proliferation in HCC cells

Firstly, we detected the expression levels of TDP-43 in different HCC cell lines by qRT-PCR and Western blot assays. The results in these cell lines showed that the expression of TDP-43 was weakest in Huh-7 cell line while abundantly in SMMC-7721 and MHCC97H cell lines (Supplementary Fig. 1a, b). Accordingly, we selected Huh-7 cells to overexpress TDP-43 and selected SMMC-7721 and MHCC97H cells to knock down TDP-43 in subsequent experiments. To explore the potential role of TDP-43 in HCC progression, we tested the effect of TDP-43 on cell proliferation by colony formation assay. Two different si-TDP-43 (termed as si-TDP-43#1 and si-TDP-43#2) were transiently transfected into SMMC-7721 and MHCC97H cells to specifically silence TDP-43 expression. The results showed that silence of TDP-43 obviously weakened the formation of colony in SMMC-7721 and MHCC97H cells (Fig. 1a, c). However, overexpression of TDP-43 in Huh-7 cells promoted the formation of colony (Fig. 1b, c). Then, we evaluated the effect of TDP-43 on apoptosis by Annexin V-FITC/PI double staining method. Silence of TDP-43 induced apoptosis in MHCC97H and SMMC-7721 cells, while TDP-43 overexpression efficiently suppressed apoptosis in Huh-7 cells (Fig. 1d–g). Next, we used MHCC97H cells to establish a TDP-43 stable-knockdown cell line (MHCC97H-KD-TDP-43) and its control cell line (MHCC97H-Control), and applied them in the subsequent studies. The expression levels of proliferation- and apoptosis-associated markers were evaluated by qRT-PCR and Western blot assay. The results displayed that TDP-43 upregulated the antiapoptotic gene *Bcl-2* and proliferation marker Ki67. Meanwhile, TDP-43 downregulated the proapoptotic genes *p53* and *Bax* (Fig. 1h, i, Supplementary Fig. 1c–e). Caspase3 is activated to form cleaved-caspase3 (C-caspase3), which is the vital terminal cleavage enzyme in apoptotic processes[22]. TDP-43 could negatively modulate the protein level of C-caspase3 (Fig. 1i). These findings demonstrate that TDP-43 prevents apoptosis and facilitates proliferation in HCC cells.

### TDP-43 accelerates growth and inhibits apoptosis in liver tumor

In order to further verify the effect of TDP-43 on cell apoptosis and proliferation in HCC, we used MHCC97H-KD-TDP-43 and MHCC97H-Control cells to conduct subcutaneous tumorigenesis experiments in nude mice. The results revealed that the weight and volume of tumors in nude mice were markedly reduced when TDP-43 was silenced (Fig. 2a–c). No difference was observed in the body weights of mice among the two groups, suggesting that the mice remained healthy upon TDP-43 knockdown (Fig. 2d). Subsequently, we evaluated the expressions of Ki67, C-caspase3, p53 and Bcl-2 in tumor tissues by immunohistochemistry (IHC) staining. IHC data uncovered that knockdown of TDP-43 depressed the expression of Ki67 and Bcl-2; Conversely, the expression levels of C-caspase3 and p53 were increased followed by TDP-43 knockdown (Fig. 2e, f). These data were in accordance with the aforesaid data in vitro, further indicating the role of TDP-43 on cell apoptosis and proliferation. In addition, we used an online resource to analyze the association between TDP-43 and the prognosis of HCC patients. The results revealed that the higher the expression level of TDP-43, the shorter the overall survival and relapse-free survival (Fig. 2g, h). Meanwhile, analysis of GSE36376 indicated that the expression level of TDP-43 in HCC tissues was substantially higher than that in adjacent non-tumor tissues (Fig. 2i). Therefore, TDP-43 is strongly associated with the poor prognosis of HCC.

### Lipid metabolism-related gene *ABHD2* is identified as a target of TDP-43

To identify the downstream target genes of TDP-43, we performed high-throughput transcriptome sequencing (RNA-seq) and obtained the transcriptome expression profiles of MHCC97H-KD-TDP-43 cells and control cells. After knocking

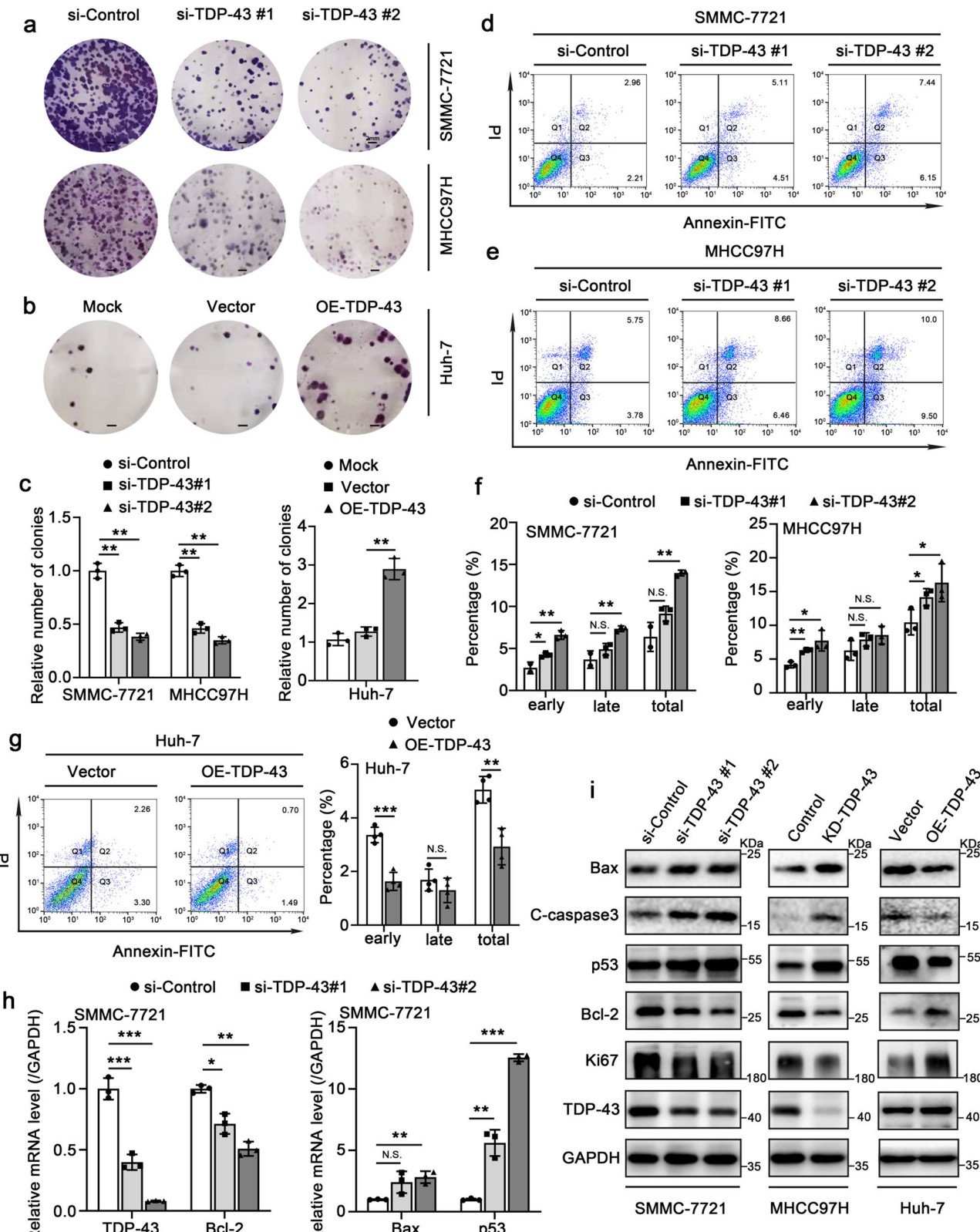

down TDP-43, the expressions of 1392 genes were increased and 1418 genes were decreased (Supplementary Fig. 2). The differentially expressed genes were displayed by a volcano plot and a hierarchical clustering heatmap (TOP100 of down-regulated genes) (Fig. 3a, b). GSEA analysis revealed that apoptosis was strongly responsive to the change in TDP-43 expression (Fig. 3c), suggesting that TDP-43 is closely related to apoptosis. This was

consistent with our previous experimental results (Figs.1 and 2). KEGG pathway enrichment analysis was performed on the down-regulated genes, and the results showed that these genes might be more involved in metabolic pathways (Fig. 3d), especially in lipid metabolism pathways (Fig. 3e). To further confirm whether TDP-43 was related to metabolism, we screened all metabolism-related genes in down-regulated genes (TOP50) for expression

**Fig. 1 TDP-43 inhibits apoptosis and promotes proliferation in HCC cells.** Colony photographs (**a**, **b**) and colony formation efficiency (**c**) after transiently transfected with the displayed siRNAs or plasmids in SMMC-7721, MHCC97H and Huh-7 cells. Scale bar: 2 mm. Flow cytometry analysis of apoptosis in SMMC-7721 (**d**) or MHCC97H (**e**) cells after transient transfection with the displayed siRNAs for 48 h and the statistical graphs (including early, late and total apoptosis rates) (**f**). **g** Flow cytometry analysis of apoptosis in Huh-7 cells after transient transfection with the indicated plasmids for 48 h (left). The right panel is the statistical graphs (including early, late and total apoptosis rates). **h** qRT-PCR analysis for the mRNA levels of *TDP-43*, *Bcl-2*, *Bax* and *p53*. SMMC-7721 cells were transiently transfected with si-Control or si-TDP-43s for 48 h. The data was normalized to the mRNA level of *GAPDH*. **i** Western blot analysis for the protein level of Bax, C-caspase3, p53, Bcl-2, Ki67 and TDP-43. The SMMC-7721 and Huh-7 cells were transiently transfected with the displayed siRNAs or plasmids. MHCC97H-KD-TDP-43 and MHCC97H-Control cells were directly used to perform Western blot analysis. The values in the graphs represent the mean of three biologically independent experiments. Error bars represent ± s. d. *$P < 0.05$, **$P < 0.01$, ***$P < 0.001$ by two-tailed Student's $t$-test.

verification. Interestingly, compared with other metabolism-related genes, the lipid metabolism-related gene, *abhydrolase domain containing 2 (ABHD2)* was the most considerably down-regulated after TDP-43 knockdown (Fig. 3f). This evidence was consistent with the analysis of the metabolic pathway in Fig. 3e. Thus, we get the conclusion that the lipid metabolism-related gene *ABHD2* is identified as a downstream target of TDP-43.

**TDP-43 is positively related to ABHD2 and increases ABHD2 expression in HCC.** Next, we performed qRT-PCR and Western blot assay in different HCC cell lines to confirm the effect of TDP-43 on ABHD2 expression. Silence of TDP-43 efficiently decreased the expression of ABHD2, while TDP-43 over-expression obviously increased ABHD2 (Fig. 4a–d and Supplementary Fig. 3a, b). Meanwhile, the expression levels of ABHD2 in mice tumor tissues were determined by IHC staining assay. We found that TDP-43 deficiency could reduce the expression of ABHD2 in tumor tissues (Fig. 4e). Moreover, the gene correlation analysis from the online database GEPIA showed that the expression of TDP-43 and ABHD2 in HCC was highly correlated (Fig. 4f, g). To further verify the correlation between the expression of TDP-43 and ABHD2 in HCC tissues, we performed IHC staining on two consecutive sections of HCC tissue micro-array. There were 32 cases of HCC tissues and paired non-tumor tissues in the tissue microarray. The result manifested that TDP-43 was positively related to ABHD2 in HCC tissues (Fig. 4h, i, and Supplementary Fig. 3c, Pearson's chi-square independent test, $\chi^2 = 61.41$, $P < 0.0001$).

**TDP-43 upregulates ABHD2 through enhancing the mRNA stability of *ABHD2*.** Next, we aimed to elucidate the upregulation mechanism of ABHD2 mediated by TDP-43. TDP-43 is an RNA-binding protein, which can regulate the splicing, transport and stability of RNA[5]. Therefore, we wondered whether TDP-43 exerted a regulatory role by binding to *ABHD2* mRNA. The RNA immunoprecipitation (RIP) assay in MHCC97H cells showed that TDP-43 could indeed bind to *ABHD2* mRNA (Fig. 5a). To further explore whether TDP-43 regulates ABHD2 at the transcriptional or post-transcriptional level, we performed mRNA stability assays in different HCC cell lines. QRT-PCR results indicated that knockdown of TDP-43 could still downregulate the mRNA level of *ABHD2* after time-dependently treated with transcription inhibitor Actinomycin-D (ActD) (Fig. 5b, c). This evidence uncovered the promotion of TDP-43 on *ABHD2* mRNA stability.

Studies have shown that TDP-43 has a high affinity for UG-rich sequences, which can bind to them and regulate the maturation of pre-mRNA[23,24]. After sequence analysis, we found there were two UG-rich sequences in the 3′UTR of *ABHD2* mRNA. Therefore, we cloned the 3′UTR of *ABHD2* mRNA and constructed it into a luciferase vector (pGL3-Control). The data showed that overexpression of TDP-43 could upregulate the luciferase activity of *ABHD2* 3′UTR, while TDP-43 silencing markedly inhibited the luciferase activity (Fig. 5d, e). Then, we

separately deleted the two UG rich sequences in *ABHD2* 3′UTR (termed as Del-1 and Del-2, respectively; the former construction of *ABHD2* 3′UTR termed as WT) (Fig. 5f), and detected the activities of luciferase reporter. After overexpression of TDP-43, the luciferase activities of WT and Del-2 were increased; similarly, the luciferase activities of WT and Del-2 were decreased upon TDP-43 deletion (Fig. 5g, h). However, the luciferase activity of Del-1 did not change obviously no matter TDP-43 was knocked down or overexpressed (Fig. 5g, h). These above results indicate that TDP-43 binds to the UG-rich sequence 1 of *ABHD2* 3′UTR, enhances the stability of *ABHD2* mRNA, and then upregulates its expression.

**Lipid metabolism modulator ABHD2 contributes to the malignant behaviors of HCC.** ABHD2, a member of α/β-hydrolase domain (ABHD) family, has the lipase serine nucleophilic motif GXSXG and can participate in lipid metabolism by exerting triacylglycerol lipase activity[25]. Under the action of triacylglycerol lipase, triacylglycerols (TG) are hydrolyzed to produce fatty acids[26,27]. Therefore, we examined the effects of ABHD2 on intracellular TG levels. The data displayed that deletion of ABHD2 obviously induced the accumulation of TG in SMMC-7721 and MHCC97H cells (Supplementary Fig. 4a, b). Meanwhile, the content of free fatty acid (FFA) was efficiently decreased after ABHD2 knockdown (Fig. 6a), indicating that ABHD2 can indeed serve as triacylglycerol lipase to promote the generation of fatty acid.

In the liver, mitochondrial β-oxidation is one of the major metabolic fates of FFAs, which will increase the production of ROS[26,28]. Studies have shown that ROS is critical for cell survival, and small increases in ROS levels can lead to the activation of some proto-oncogenic pathways and facilitate cancer development[29,30]. We examined the cellular ROS level and found that ABHD2 could indeed induce the generation of ROS (Fig. 6b, c). Thus, we speculated that ABHD2 might affect the development of HCC through the ROS produced by FFA metabolism. Expectedly, the silence of ABHD2 remarkably increased the apoptotic rate in SMMC-7721 and MHCC97H cells (Fig. 6d–f). The result of western blot further showed that after ABHD2 knocking down, the expression levels of p53, C-caspase3 and Bax were elevated, while the expression level of antiapoptotic gene Bcl-2 was reduced (Fig. 6g and Supplementary Fig. 4c). In addition, CCK-8 and colony formation assay showed that ABHD2 deficiency also resulted in the inhibition of cell proliferation (Supplementary Fig. 4d, e). Then, we aimed to investigate the role of ROS in ABHD2-suppressed apoptosis. We carried out flow cytometry, qRT-PCR, and Western blot assays in ABHD2-overexpressed Huh-7 cells upon the treatment of ROS scavenger N-Acetyl-L-cysteine (NAC). The results showed that NAC treatment could alleviate the inhibition of apoptosis induced by ABHD2 overexpression (Supplementary Fig. 5), suggesting that ABHD2 inhibits apoptosis in an ROS-dependent manner.

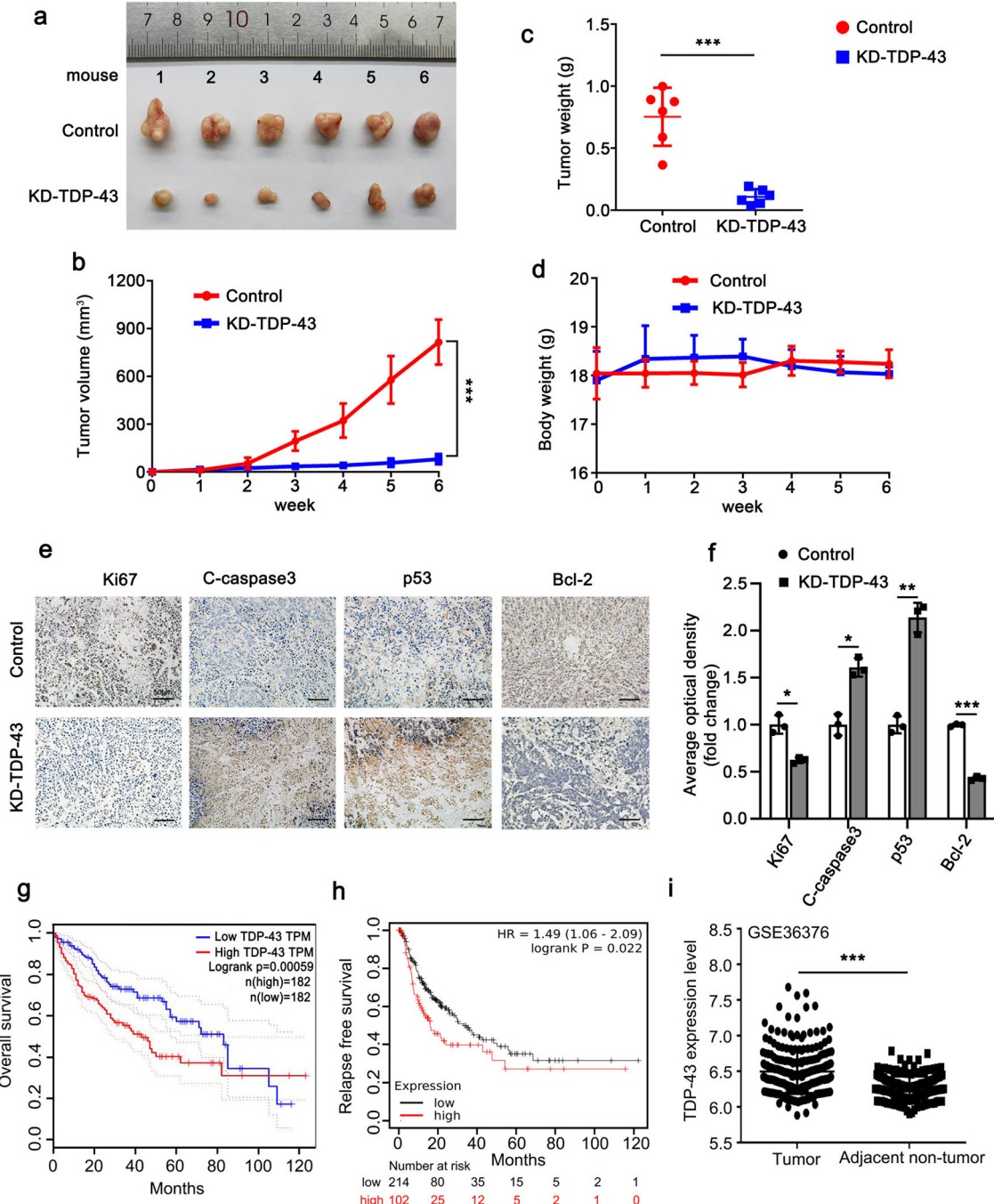

**Fig. 2 TDP-43 accelerates growth and inhibits apoptosis in liver tumor.** Tumor images (**a**), tumor growth curves (**b**), and tumor weights (**c**) of the xenograft tumors derived from MHCC97H-KD-TDP-43 and MHCC97H-Control cells ($n = 6$). **d** The body weights of mice in the indicated groups. IHC staining of Ki67, C-caspase3, p53, and Bcl-2 in mice xenograft tumors (**e**) and quantitative analysis (**f**). Scale bar: 50 μm. **g, h** Overall survival and relapse-free survival were analyzed in HCC cancer patients using the GEPIA (http://gepia.cancer-pku.cn/) and Kaplan–Meier plotter (http://kmplot.com/analysis/) online databases, respectively. Plots were generated based on the expression level of TDP-43 (g plot: logrank $P = 0.00059$; h plot: logrank $P = 0.022$). **i** The expression levels of TDP-43 in liver tumor tissues and normal tissues adjacent to the tumor were counted from GEO database GSE36376. Error bars represent ± s. d. *$P < 0.05$, **$P < 0.01$, ***$P < 0.001$ by two-tailed Student's $t$ test.

**TDP-43 facilitates lipid metabolism via ABHD2 to suppress apoptosis in vitro.** Next, we wondered whether TDP-43 could modulate lipid metabolism by ABHD2 to suppress apoptosis. First, we performed TG assays in SMMC-7721 and MHCC97H cells. Notably, TDP-43 knockdown induced the accumulation of TG while elevation of ABHD2 could alleviate this phenomenon (Supplementary Fig. 6a). Furthermore, when TDP-43 was silenced, the FFA content was decreased (Fig. 7a). However,

overexpression of ABHD2 could efficiently relieve the decrease of FFA induced by TDP-43 deficiency (Fig. 7a). Determination of ROS level also manifested the rescue effect of ABHD on TDP-43 deletion-induced decrease of ROS (Fig. 7b, c). Then, we aimed to confirm the role of ABHD2 in TDP-43-regulated cell apoptosis. Expectedly, ABHD2 markedly suppressed the apoptosis induced by the lack of TDP-43 (Fig. 7d, e). QRT-PCR and western blot further showed that ABHD2 disrupted the upregulation of p53

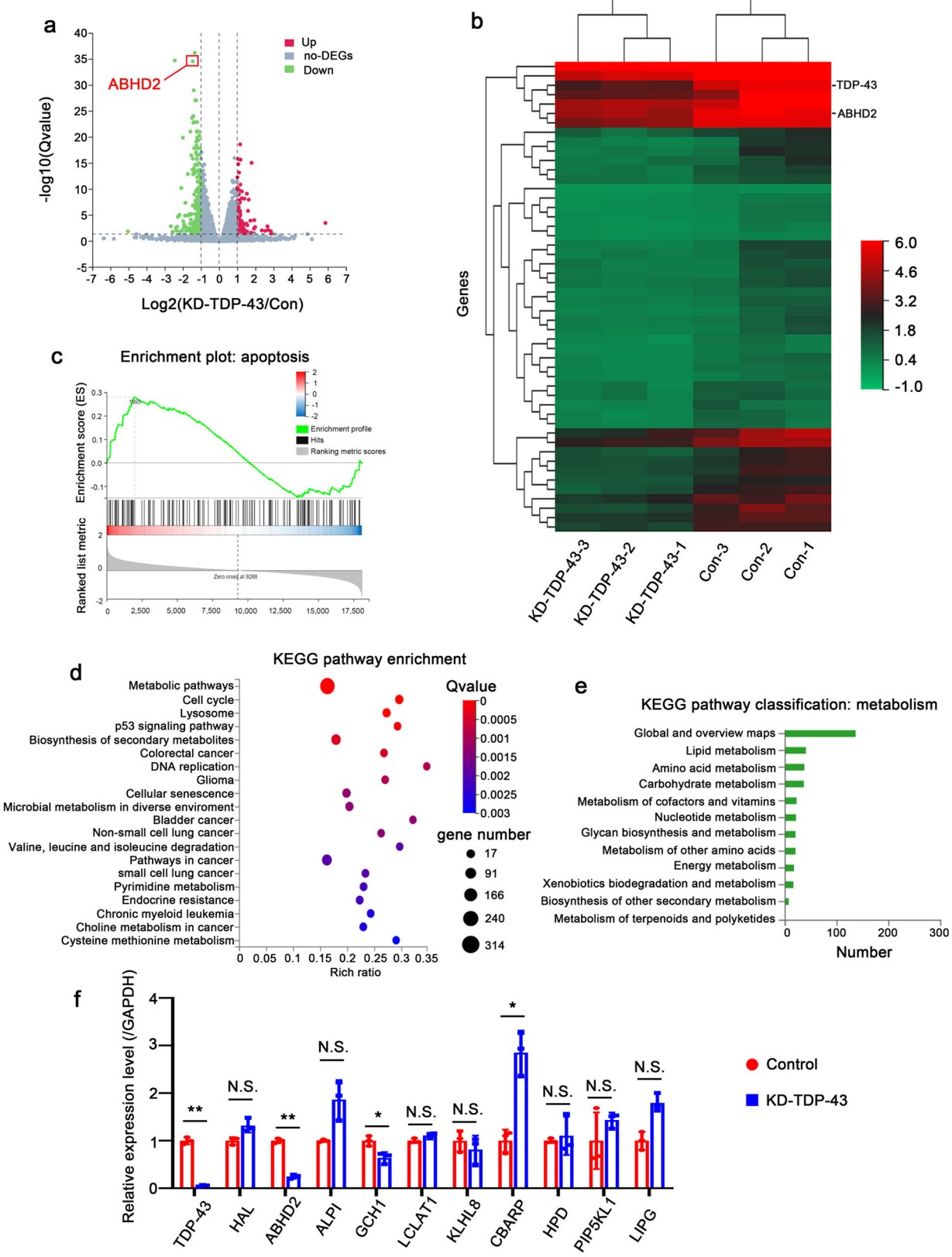

and C-caspase3 evoked by TDP-43 knockdown; TDP-43 deletion induced-downregulation of Bcl-2 and Ki67 was also improved by ABHD2 (Fig. 7f, g and Supplementary Fig. 6b). In addition, CCK-8 data uncovered that elevation of ABHD2 could improve the decline of cell proliferation induced by TDP-43 knockdown (Supplementary Fig. 6c). Considering that the anti-apoptotic

effect of TDP-43 was mediated through ABHD2-associated lipid metabolism, we next evaluated the role of the metabolite FFA in TDP-43-inhibited apoptosis. We performed rescue experiments by FFA in SMMC-7721 cells with knockdown of TDP-43. Flow cytometry, qRT-PCR and Western blot assays showed that FFA treatment effectively reversed the apoptosis caused by TDP-43

**Fig. 3 Lipid metabolism-related gene ABHD2 is identified as a target of TDP-43. a** Volcano map of differentially expressed genes in HCC cells upon TDP-43 knockdown. Red dots, genes drastically up-regulated. Green dots, genes drastically down-regulated. Gray dots, genes with no considerable difference in expression. **b** Cluster heat map of downregulated genes (TOP100) in HCC cells after knockdown of TDP-43. **c** GSEA plot of the apoptosis pathway in RNA-Seq data upon silencing of TDP-43 in MHCC97H cells. **d** Bubble plot of KEGG pathway enrichment data. The horizontal coordinate is the rich ratio, and the larger value indicates the greater enrichment. Q value is the multiple-checked *p* value, and the redder value indicates the smaller Q value, indicating the more obvious enrichment. **e** Histogram of KEGG metabolism-related pathway gene enrichment. **f** qRT-PCR assays for the 10 metabolism-associated genes among the Top 50 differentially expressed genes. MHCC97H-KD-TDP-43 and MHCC97H-Control cells were used in this experiment. The data was normalized to the mRNA level of *GAPDH*. The values in the graphs represent the mean of three biologically independent experiments. Error bars represent ± s. d. *$P < 0.05$, **$P < 0.01$, ***$P < 0.001$ by two-tailed Student's *t* test.

deletion (Supplementary Fig. 6d–g). These findings indicate that TDP-43 is able to suppress apoptosis through ABHD2-mediated lipid metabolism.

**TDP-43 facilitates lipid metabolism via ABHD2 to suppress apoptosis in vivo.** Next, we constructed stable overexpression of ABHD2 cell lines based on MHCC97H-KD-TDP-43 and MHCC97H-Control cells. The above cells (MHCC97H-Control + vector, MHCC97H-KD-TDP-43 + vector and MHCC97H-KD-TDP-43 + ABHD2) were collected for subcutaneous tumor implantation in nude mice. After 42 days, the subcutaneous tumors of three groups were collected and the tumor weight was detected. The results showed that overexpression of ABHD2 effectively alleviated the inhibition of tumor growth caused by TDP-43 deletion (Fig. 8a–c). Examination of FFA content in mice tumors also manifested the same result: TDP-43 deletion induced-reduction of FFA could be rescued by ABHD2 overexpression (Fig. 8d). Then, the protein abundances of TDP-43, ABHD2, Ki67, C-caspase3, p53 and Bcl-2 in tumor tissues were detected by IHC staining. We observed that deletion of TDP-43 caused the downregulation of Ki67 and Bcl-2, and the upregulation of p53 and C-caspase3 in mice tumors in contrast to the control group (Fig. 8e, f). However, ABHD2 overexpression appreciably relieved the above phenomenon (Fig. 8e, f). Taken together, these findings disclose that TDP-43 can facilitate lipid metabolism by upregulating ABHD2, to inhibit apoptosis in HCC cells.

## Discussion

Over the years, with the development of medicine, we have made considerable progress in the diagnosis and treatment of HCC; However, statistics have showed that almost 70% of patients recur within 5 years after operation[31]. Therefore, the exploration of strategies and targets for the treatment of HCC has never stopped. The poor prognosis of HCC is mainly related to the delay of diagnosis[32]. Study of biomarkers may provide more instructive information for the individualized diagnosis and/or prognosis of HCC. The role of TDP-43 in neurological diseases has been well studied, but its potential function in malignant tumors still needs to be explored. In our study, survival analysis manifested that the higher the expression level of TDP-43, the shorter the survival rate of HCC patients. Moreover, TDP-43 had been proved to inhibit apoptosis and promote proliferation in HCC cells, thus accelerating the malignant development of HCC. Our discovery is expected to provide a biomarker for the treatment of HCC, so as to be applied to the diagnosis and prognosis of patients.

To elucidate the mechanism of TDP-43 promoting HCC, we implemented high-throughput sequencing and found that TDP-43 was indeed closely related to the apoptotic pathway. Interestingly, KEGG analysis showed that metabolic pathways strongly responded to the changes in TDP-43 expression. Therefore, we suspected that TDP-43 was likely to suppress the apoptosis of HCC by affecting metabolic pathways. The further classification analysis of KEGG pathway discovered the close association

between lipid metabolism and TDP-43. Lipids are critical to the cellular life as molecules for energy storage, membrane structure and signal transduction[33,34]. So, we supposed whether TDP-43 modulated lipid metabolism-related gene in HCC. In order to confirm our speculation, we detected the expression of all metabolism-related genes among the differential downregulated genes (Top50) and observed that the lipid metabolism-related gene *ABHD2* was most dramatically downregulated after TDP-43 silencing. ABHD2, as a member of the mammalian ABHD family, is a triacylglycerol lipase with ester hydrolysis ability[25]. Conserved structural motifs shared by the members of ABHD family primarily play a role in lipid synthesis and degradation. In recent years, the ABHD family has emerged as a potential regulator of signal transduction and lipid metabolism[34]. The above findings led us to investigate whether TDP-43 could modulate lipid metabolism *via* ABHD2. TG and FFA content examination displayed that TDP-43 enhanced the TG-originated production of FFA in an ABHD2-dependent manner. Fatty acid oxidation (FAO)-originated ROS maybe a major source of ROS and plays a crucial role in oxidative stress during FFA overload in the liver[35,36]. By detecting intracellular ROS levels, we discovered TDP-43 was able to elevate the content of ROS through ABHD2. The production of ROS could activate multiple oncogenic signaling pathways and furthermore contribute to the pathogenesis of multiple diseases, including cancers[37,38]. Expectedly, we found that ABHD2 could robustly attenuate the promotion of apoptosis induced by TDP-43 deletion. Moreover, FFA treatment also efficiently alleviate TDP-43 deficiency-caused apoptosis. These evidences revealed that with the help of ABHD2, TDP-43 regulated lipid metabolism, which had an impressive effect on tumor cell apoptosis.

Notably, few studies have addressed the role of ABHD2 in malignant tumors. It is reported that the expressions of ABHD2 in breast cancer and melanoma are obviously higher than those in normal tissues[39–41], but whether it has molecular functions in these two cancers is still unknown. The Obinata team has reported that *ABHD2* is an androgen-responsive gene, and it can promote the growth and migration of prostate cancer[42]. However, in HCC, the expression and underlying function of ABHD2 remains to be elucidated. In this study, we first put forward that ABHD2 played a crucial role in inhibiting apoptosis and promoting proliferation in HCC. A study in 2009 found that increased apoptosis could be observed in *ABHD2* gene deficient mice[43], which was consistent with our findings. In conclusion, our data suggested that ABHD2 had significance in tumor development and was expected to be a potential target for the precise treatment of HCC.

In the study of the mechanism of TDP-43 regulating ABHD2, we confirmed that TDP-43 bind to the site1 (UG-rich sequence1) to enhance the mRNA stability of *ABHD2*, thus upregulating ABHD2. Interestingly, we also found that ABHD2 had a certain effect on the expression of TDP-43. Western blot and qRT-PCR data displayed that the level of TDP-43 expression was upregulated after ABHD2 overexpression (Fig. 7f, g). Especially in the

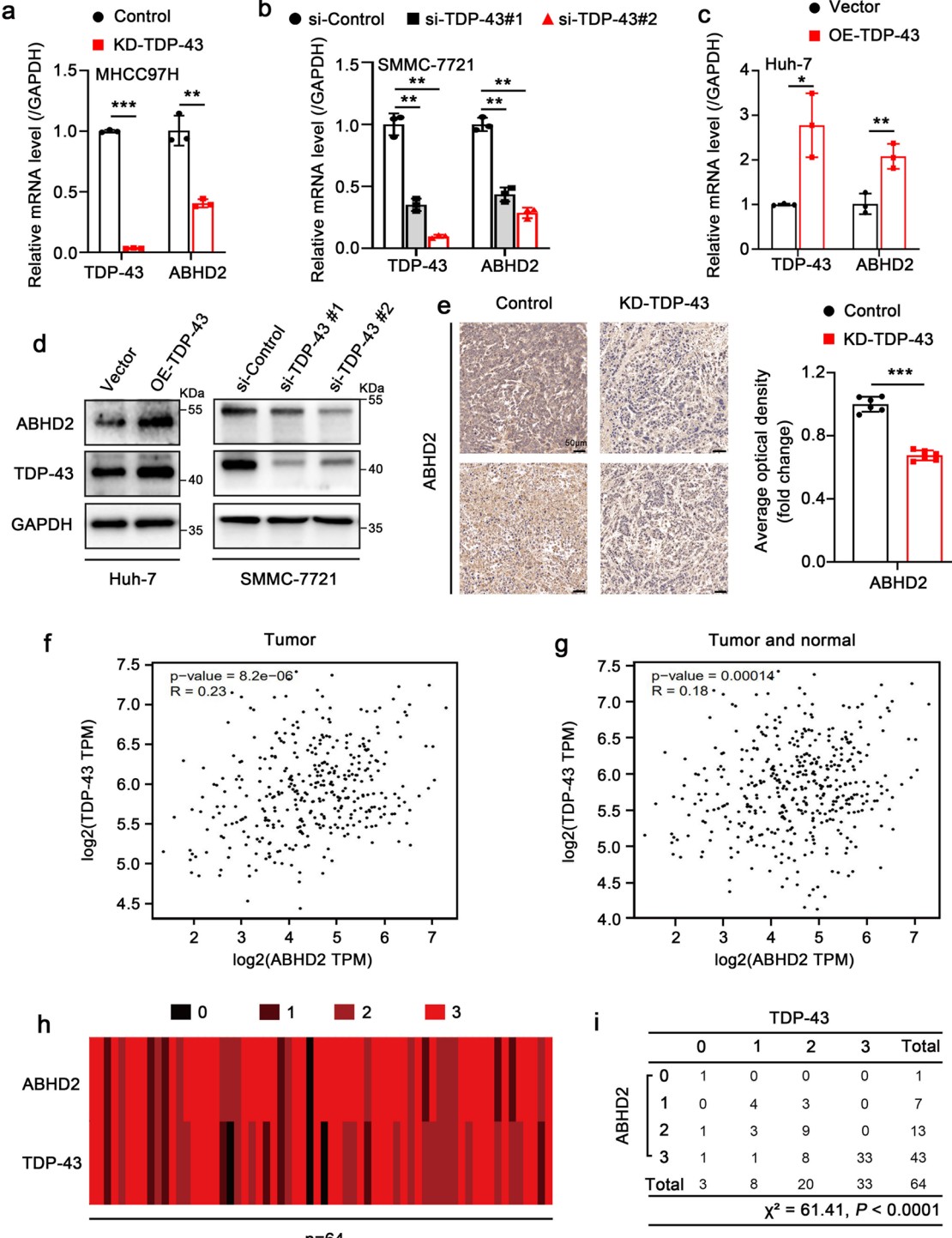

**Fig. 4 TDP-43 is positively related to ABHD2 and increases ABHD2 expression in HCC. a–c** qRT-PCR assay to detect the effect of TDP-43 on ABHD2 expression in HCC cells. The MHCC97H-KD-TDP-43 and MHCC97H-Control cells were used for the assay in (**a**). The cells (SMMC-7721 and Huh-7) were transiently transfected with the indicated siRNAs or plasmids. The data was normalized to the mRNA level of GAPDH. **d** Western blot analysis was performed to detect the effect of TDP-43 on ABHD2 expression in Huh-7 and SMMC-7721 cells. The cells were transiently transfected with the indicated siRNAs or plasmids for 48 h. **e** IHC staining of ABHD2 in the xenograft tumors derived from MHCC97H-KD-TDP-43 and MHCC97H-Control cells. The right panel is the quantitative analysis. Scale bar: 50 μm. Expression correlation analysis of *TDP-43* and *ABHD2* genes in Tumor (**f**), and Tumor and normal (**g**) in the GEPIA database (http://gepia.cancer-pku.cn/detail.php). **h** Heat map of the expression levels of TDP-43 and ABHD2 in HCC tissue microarray. Numbers 0, 1, 2 or 3 represent negative, weak, moderate or strong staining, respectively. **i** Statistical analysis of the association between TDP-43 and ABHD2 expression levels in the abovementioned tissue microarray by Pearson chi-square independent test, $\chi^2 = 61.41$, $P < 0.0001$. The values in the graphs represent the mean of three biologically independent experiments. Error bars represent ± s. d. *$P < 0.05$, **$P < 0.01$, ***$P < 0.001$ by two-tailed Student's $t$ test.

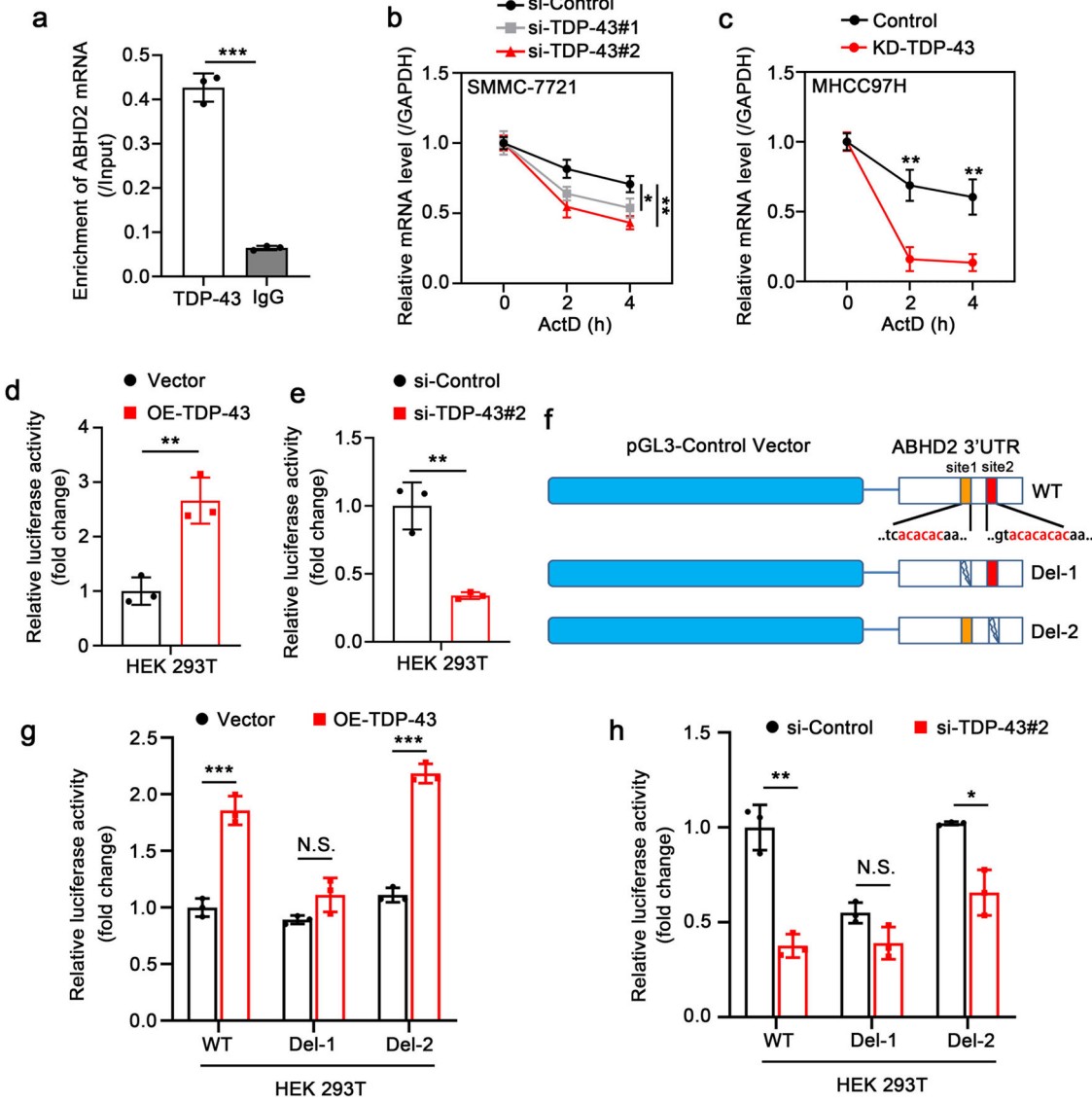

**Fig. 5 TDP-43 upregulates ABHD2 through enhancing the mRNA stability of ABHD2. a** RIP assay to detect the interaction between TDP-43 and *ABHD2* mRNA in MHCC97H cells. The data was normalized to the Input. **b** The mRNA stability of *ABHD2* was analyzed by qRT-PCR in SMMC-7721 cells upon TDP-43 deletion. The cells were treated with ActD for the indicated time points after transient transfection for the indicated siRNAs. **c** The mRNA stability of *ABHD2* was analyzed by qRT-PCR in MHCC97H-KD-TDP-43 and MHCC97H-Control cells. The cells were treated with ActD for the indicated time points. Luciferase reporter gene assay of *ABHD2* 3′UTR activity upon TDP-43 overexpression (**d**) or knockdown (**e**). The cells were transiently transfected with the indicated plasmids or siRNAs. **f** Schematic representation of *ABHD2* 3′UTR constructed on pGL3-Control plasmid. WT represents the wild-type plasmid of *ABHD2* 3′UTR; Del-1 represents the WT plasmid without the binding site1 (UG-rich sequence1, 6704–6709 refer to NM_007011.8); Del-2 represents the WT plasmid without the binding site2 (UG-rich sequence2, 7122–7129 refer to NM_007011.8). Luciferase reporter gene assay of *ABHD2* 3′UTR activities upon TDP-43 overexpression (**g**) or knockdown (**h**). The HEK 293 T cells were transiently transfected with WT, Del-1 or Del-2 accompanied with the indicated plasmids or siRNAs. The values in the graphs represent the mean of three biologically independent experiments. Error bars represent ± s. d. *$P < 0.05$, **$P < 0.01$, ***$P < 0.001$ by two-tailed Student's *t* test.

qRT-PCR experiment, ABHD2-induced upregulation of TDP-43 showed noticeable statistical significance. Therefore, we speculated that ABHD2 might be able to reversely regulate the expression of TDP-43. However, whether there was a positive feedback loop between TDP-43 and ABHD2 to jointly promote the development of HCC or other tumors is still unclear.

In summary, our study discloses a mechanism for TDP-43-promoted HCC progression (Fig. 9). As an RNA-binding protein, TDP-43 can bind to the mRNA 3′UTR of lipid metabolism-related gene *ABHD2*, enhance the stability of *ABHD2* mRNA, and then upregulate ABHD2. Elevated-ABHD2 catalyzes the metabolic process of TG to FFAs, thus promoting the β-oxidation

pathway and increasing the production of ROS, finally resulting in the suppression of apoptosis and facilitation of proliferation in HCC. Our study provides a theoretical basis for elucidating the role of RNA-binding proteins in the promotion of malignance. Meanwhile, TDP-43/ABHD2 can be used as potential target, which has important application value in the precise treatment of HCC.

## Methods

**Plasmid construction and small interference RNA (siRNA).** Using the mRNA extracted from SMMC-7721 cells as the template, the coding region sequence (CDS) of *TDP-43* was amplified by RT-PCR. During amplification, the enzyme

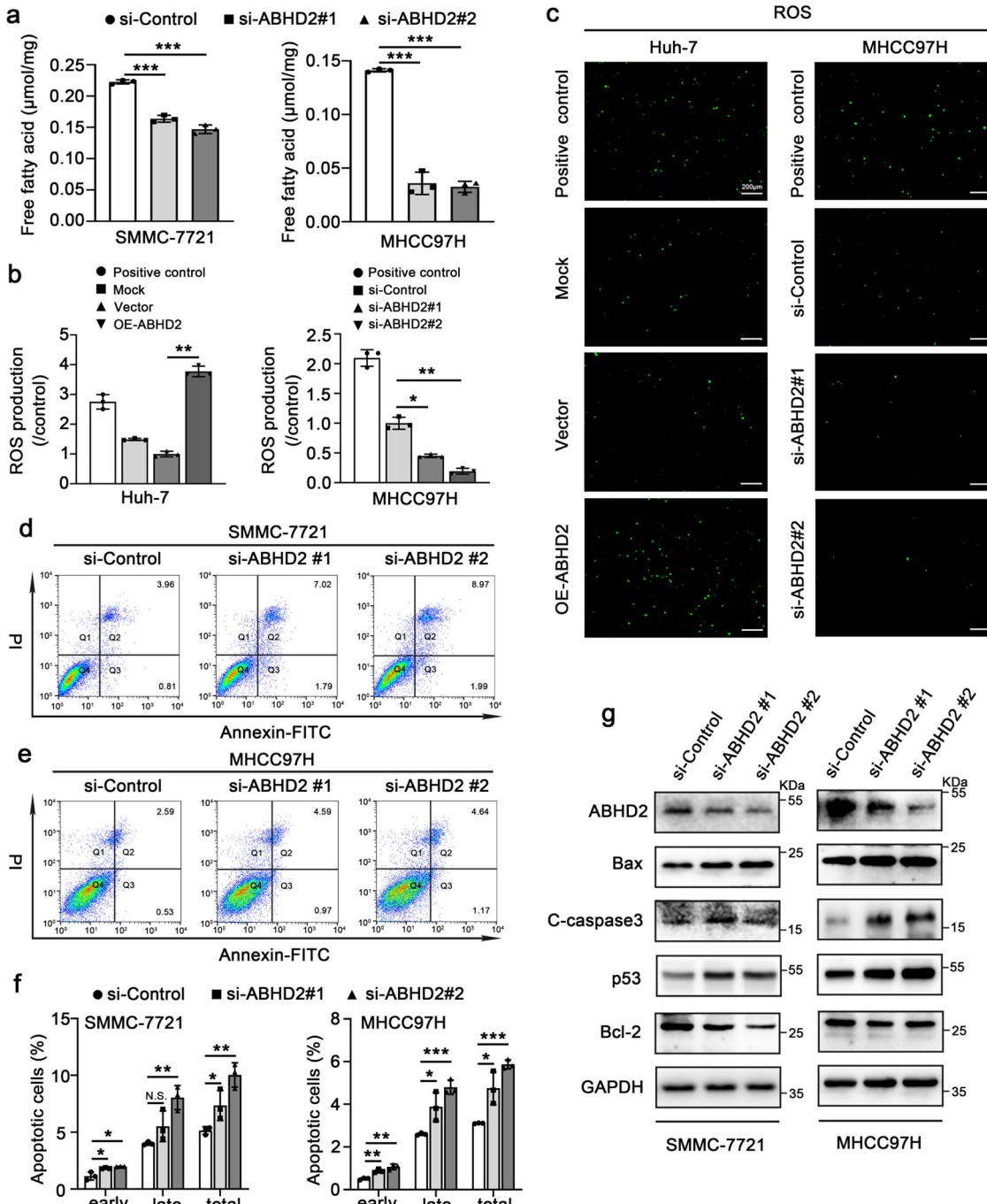

**Fig. 6 Lipid metabolism modulator ABHD2 contributes to the malignant behaviors of HCC. a** Analysis of free fatty acid (FFA) levels in SMMC-7721 (left) and MHCC97H (right) cells which were transiently transfected with si-Control or si-ABHD2s. Quantitative analysis (**b**) and fluorescence images (**c**) of reactive oxygen species (ROS) in Huh-7 and MHCC97H cells transiently transfected with the displayed plasmid or siRNAs. Scale bar: 200 μm. **d–f** Flow cytometry analysis of apoptosis in SMMC-7721 and MHCC97H cells transfected with si-Control or si-ABHD2s for 48 h. The statistical graphs of apoptotic rate were shown in (**f**). **g** Western blot analysis of protein levels of ABHD2, Bax, C-caspase3, p53, and Bcl-2 in SMMC-7721 and MHCC97H cells transiently transfected with si-ABHD2s or si-Control. The values in the graphs represent the mean of three biologically independent experiments. Error bars represent ± s. d. *P < 0.05, **P < 0.01, ***P < 0.001 by two-tailed Student's t-test.

digestion sites were introduced. The PCR products and vector pEGFP-C2 were double digested, respectively. Then, the PCR products were connected with pEGFP-C2 by T4 DNA ligase. Finally, the recombinant plasmid pEGFP-TDP-43 was used for exogenous overexpression of TDP-43. The pCMV3-ABHD2 expression plasmid was purchased form Sino Biological (China, HG14770-CF) and was used for ABHD2 overexpression. The 3′UTR of *ABHD2* was amplified by PCR using specific primers and inserted into the pGL3-Control vector to generate the pGL3-ABHD2-3′UTR (WT). Then, two sites rich in UG sequence were separately deleted to construct two different *ABHD2* 3′UTR mutants (Del-1 and Del-2). All siRNAs used in this study were generated by RiboBio (Guangzhou, China), and the

sequences of siRNAs were listed in the Supplementary Table 1. The primer sequences used in ABHD2 3′UTR construction were shown in Supplementary Table 3.

**Western blot assay**. Total proteins were extracted by RIPA buffer (Beyotime, China) containing 1nmol/L PMSF (Beyotime). The proteins were resolved by SDS-PAGE and then transferred on a polyvinylidene difluoride membrane (GE Healthcare Life Sciences, USA). After incubation with 5% skimmed milk at room temperature for 2 h, the blot was hybridized with the primary antibody at 4 °C

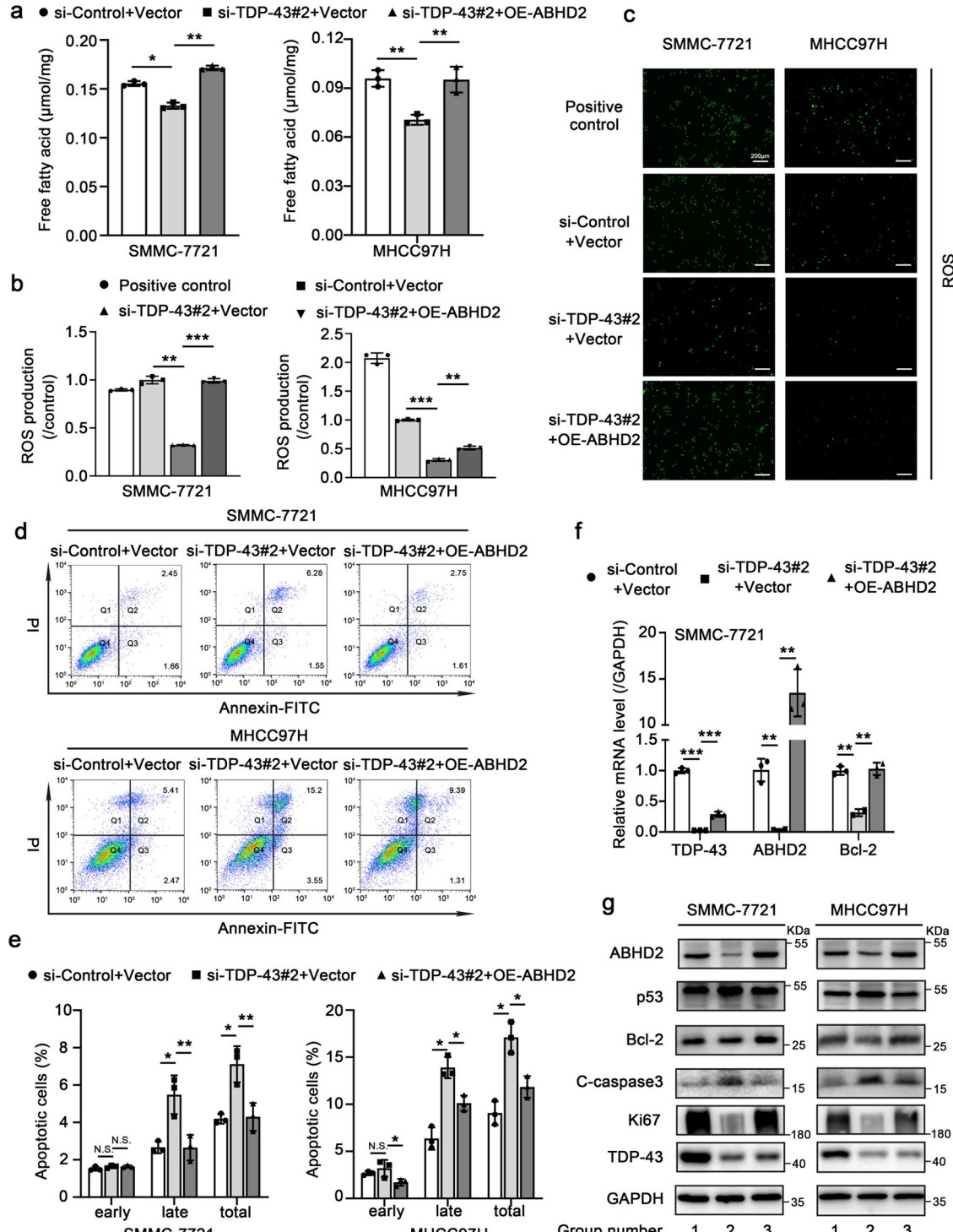

**Fig. 7 TDP-43 facilitates lipid metabolism via ABHD2 to suppress apoptosis in vitro. a** Analysis of FFA levels in SMMC-7721 (left) and MHCC97H (right) cells. The cells were transiently transfected with si-Control/si-TDP-43#2 and pCMV3 (vector)/pCMV3-ABHD2 to silence TDP-43 or overexpress ABHD2. Quantitative analysis (**b**) and fluorescence images (**c**) of reactive oxygen species (ROS) in SMMC-7721 and MHCC97H cells after transient transfection with the displayed siRNAs or plasmids. Scale bar: 200 μm. **d** Flow cytometry analysis of apoptosis in SMMC-7721 (upper panel) and MHCC97H (lower panel) cells upon TDP-43 deletion along with or without ABHD2 overexpression. **e** Quantitative analysis of apoptotic rate in (**d**). **f** qRT-PCR assay for the mRNA levels of *TDP-43, ABHD2* and *Bcl-2* in SMMC-7721 cells upon TDP-43 deletion along with or without ABHD2 overexpression. The data was normalized to the mRNA level of *GAPDH*. **g** Western blot analysis of protein levels of ABHD2, p53, Bcl-2, C-caspase3, Ki67, and TDP-43 in SMMC-7721 and MHCC97H cells. Group number 1, 2, and 3 separately represents the three groups shown in (**f**). The cells were transiently transfected with the displayed siRNAs or plasmids. The values in the graphs represent the mean of three biologically independent experiments. Error bars represent ± s. d. *$P < 0.05$, **$P < 0.01$, ***$P < 0.001$ by two-tailed Student's $t$ test.

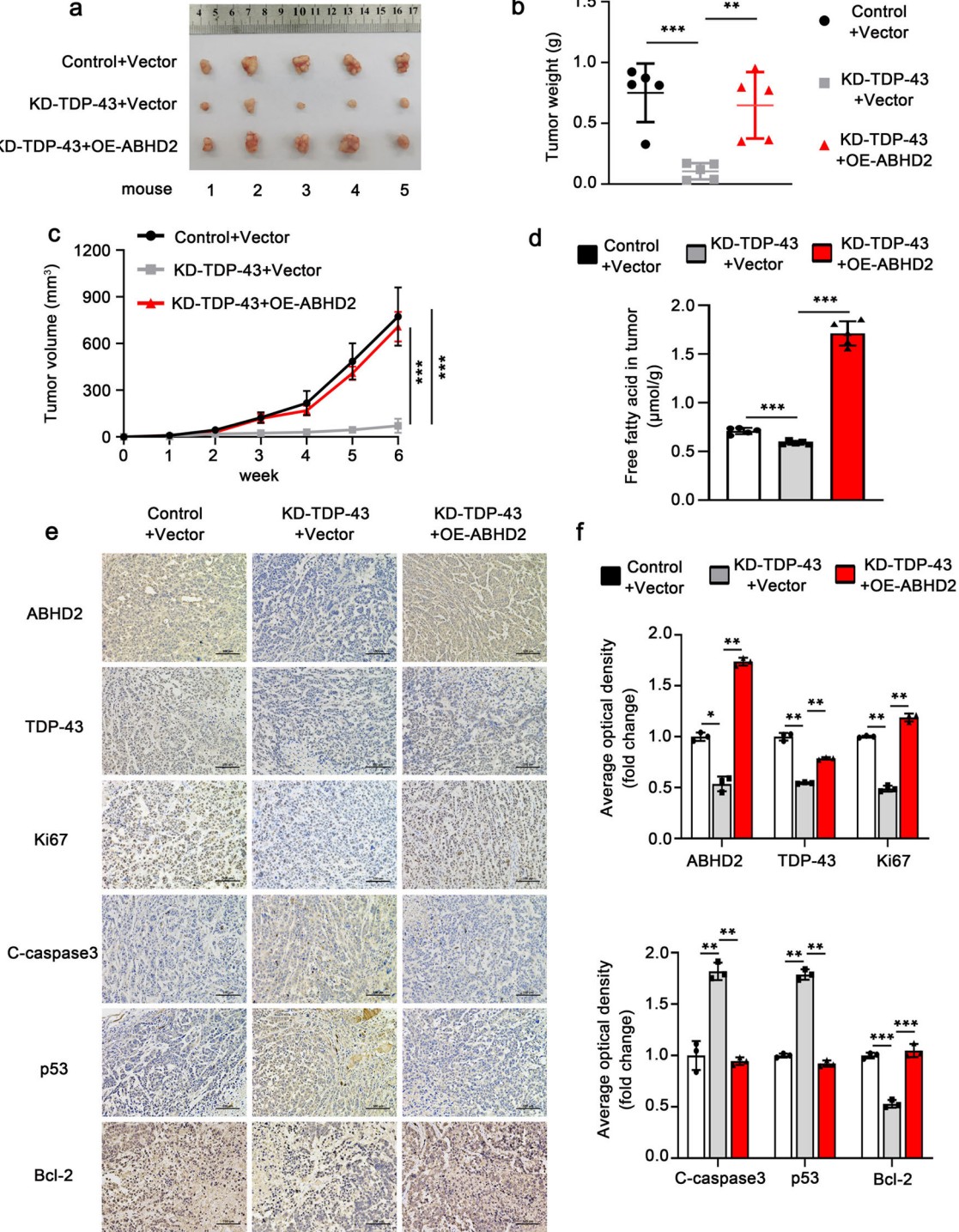

**Fig. 8 TDP-43 facilitates lipid metabolism via ABHD2 to suppress apoptosis in vivo.** Tumor images (**a**), tumor weights (**b**), and tumor growth curves (**c**) of the xenograft tumors derived from MHCC97H-Control+vector, MHCC97H-KD-TDP-43+vector and MHCC97H-KD-TDP-43+OE-ABHD2 cells ($n = 5$). **d** Analysis of FFA levels in mice tumors of the indicated three groups. IHC staining of ABHD2, TDP-43, Ki67, C-caspase3, p53 and Bcl-2 in mice tumors (**e**) and quantitative analysis (**f**). Scale bar: 100 μm. Error bars represent ± s. d. *$P < 0.05$, **$P < 0.01$, ***$P < 0.001$ by two-tailed Student's *t* test.

overnight. After washing by TBST three times, the blot was incubated with the secondary antibody (Beyotime, A0208, A0216, dilution: 1:1000) at room temperature for 1 h. After adding the enhanced chemiluminescence (NCM Biotech, China), the immunoblots were obtained using Azure600 (azure biosystems, USA). The primary antibodies used in this study were GAPDH (Abcam, UK, ab181602, dilution: 1:5000), TDP-43 (Abcam, ab190963, dilution: 1:2000), ABHD2 (absin, China, abs140798, dilution: 1:1000), Cleaved-caspase3 (Cell Signaling, USA, #9664, dilution: 1:2000), p53 (Proteintech, USA, 10442-1-AP, dilution: 1:5000), Bax (Cell Signaling, #5023, dilution: 1:1000), Bcl-2(Abcam, ab182858, dilution: 1:2000), Ki67 (Abcam, ab16667, dilution: 1:1000).

**Flow cytometry assay.** In total, $1 \times 10^5$ cells were plated into 6-well plates and were harvested after transfection for 48 h. After washing with precooled PBS, the binding buffer was added to resuspend cells. Subsequently, dye the cells with 5 μl Annexin V-FITC and 5 μl propidium (KeyGen Biotech, China). Within 1 h, the apoptotic cells were detected by flow cytometry. Three replicates were set in each group.

**Determination of FFA content.** To evaluate intracellular lipid metabolism, cells were seeded on a 6-well plate and collected after transfection for 48 h. According to the manufacturer's operating instructions, the FFA levels were determined using a

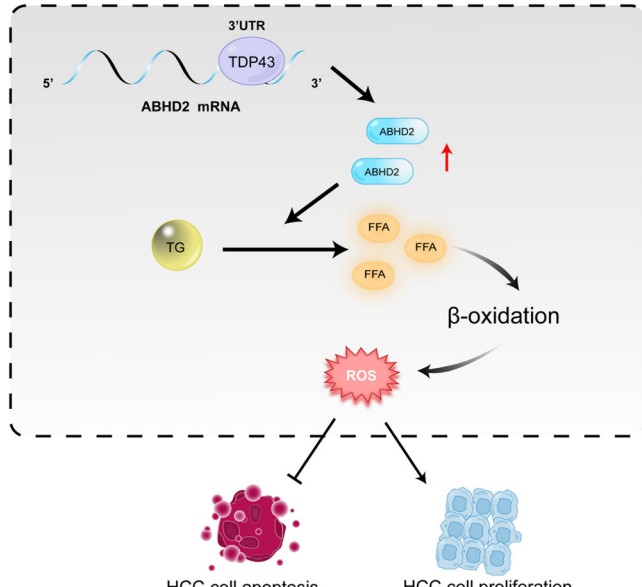

**Fig. 9 Schematic diagram of TDP-43 suppressing apoptosis by modulating ABHD2 in HCC.** TDP-43 binds with the 3′UTR of *ABHD2* mRNA to enhance its mRNA stability, thereby upregulating ABHD2 expression. Elevated-ABHD2 catalyzes the metabolic process of TG to FFAs, thus promoting the β-oxidation pathway and increasing the production of ROS, finally resulting in the suppression of apoptosis and facilitation of proliferation in HCC.

Free Fatty Acid Quantification Kit (CoMIN, China). For tumor metabolism analysis, the mice tumor tissues need to be ground in an ice bath first, and then assayed for FFA content according to the instructions.

**Immunohistochemistry (IHC) staining.** Paraffin-embedded xenograft sections were used for TDP-43 (Servicebio, China, GB112160, dilution: 1:400), ABHD2 (absin, abs140798, dilution: 1:200), C-caspase3 (Servicebio, GB11532, dilution: 1:500), Bcl-2 (Servicebio, GB113375, dilution: 1:500), and p53 (Servicebio, GB111740, dilution: 1:500) staining analysis. The antibody of Ki67 used in this experiment was purchased from Abcam (ab15580, dilution: 1:200). After deparaffinized and rehydrated, sections were laid in EDTA solution (pH9.0) for antigen retrieval by boiling in microwave oven for 8 min on medium heat. The slides were naturally cooled before treating with 3% $H_2O_2$ for 10 min. Sections were blocked in 10% normal goat serum for 30 min at room temperature, and then incubated overnight with indicated primary antibody at 4 °C. After three washes with PBS, the slides were hybridized with the secondary antibody (Servicebio, GB23303, dilution: 1:200) for 50 min at room temperature. Finally, diaminobenzidine (DAB) (servicebio) was used for color development. After counterstaining with hematoxylin, dehydrate the slides and seal with neutral gum (Solarbio, China). Image J software was used for staining analysis.

**Tissue microarray analysis.** HCC tissue microarray (No. live c-1401) was purchased from Servicebio. It contains 32 cases of HCC tissues and paired non-tumor tissues, including hepatocellular carcinoma, cholangiocarcinoma, and mixed hepatocellular carcinoma-cholangiocarcinoma with different degrees of differentiation. See Supplementary table 4 for patient records. The tissue microarray was stained by IHC, and the staining analysis was performed by Image J. The grading criteria for the staining levels of TDP-43 and ABHD2 were as follows: During microscopic observation, 5 different sights were randomly chosen for staining evaluation. The assessment was divided into two parameters: positive staining rate and staining intensity. For the staining rate, 0% stained = 0; 1–29% stained = 1; 30–65% stained = 2; 66–100% stained = 3. For the staining intensity, negative = 0; low = 1; moderate = 2; high = 3. Finally, multiply the scores of the above two parameters. When the result is less than 1, the final score is recorded as 0 (nonstaining); result is 1, 2 or 3, the final score is recorded as 1 (weak staining); result is 4 or 6, the final score is 2 (moderate staining); result is 9, the final score is 3 (intense staining). The heat map is drawn and analyzed according to the above final scores.

**Xenograft.** Beijing Vital River Laboratory Animal company provided the BALB/c Nude mice (carrying $Foxn1^{nu}$ mutation) for the xenograft experiment. The experiment followed the guidelines of the Ethics Committee of Xinxiang Medical University, and nude mice were fed according to the National Institutes of Health Guidelines for the Care and Use of Laboratory Animals. 6- to 8-week-old male mice were grouped randomly ($n = 5$ or 6). After growing to the logarithmic growth phase, the cells were collected and implanted subcutaneously in the right shoulder of each nude mouse. One week after subcutaneous inoculation, the volume of tumor was measured every three days, the weight of mice was recorded as well. Six weeks later, the nude mice were euthanized and dissected, and the tumor was photographed and weighed. Then, the tumor tissues were analyzed by immunohistochemistry and free fatty acid content in order to analyze the related proliferation, apoptosis and metabolic indexes.

**RNA-sequencing (RNA-seq).** MHCC97H-KD-TDP-43 and MHCC97H-Control cells were used as the research objects, and there were three replicates in each set. The above cells were grown to logarithmic phase and were collected to extract RNA. The concentration and purity of RNA were measured. Then, the specimen was sent to HuaDa BGI (China) for RNA-seq. BGI is responsible for the construction of mRNA library and analysis of transcriptome data. For detailed protocols, please visit the official website of BGI. RNA-seq reads were quantified as Fragments Per kilobase Million.

**RNA immunoprecipitation (RIP).** The RIP assay was conducted to examine the interaction of TDP-43 and *ABHD2* mRNA. Briefly, HCC cells were gathered by centrifugation and then lysed with RIP lysis buffer containing RNAase inhibitor. The protein A + G agarose beads (Millipore, USA) were re-suspended and incubated with TDP-43 antibody or control IgG for 1 h. Next, the cell lysate was incubated with the above-prepared agarose beads at 4 °C overnight. After repeated cleaning with NT-2 buffer for 6–8 times, added protease K buffer into agarose beads and water bath at 55 °C for 30 min to remove protein. Finally, NT-2 buffer was added and immunoprecipitated RNA was extracted with Trizol reagent for qRT-PCR analysis.

**Statistics and reproducibility.** All replicates displayed in this paper are biological replicates; technical replicates (usually three) were performed and used to generate the means for each biological replicate. The sample sizes and number of replicates were indicated in the figure legends. Graph pad prism 8.0 (San Diego, USA) was used for statistical analysis. The statistical significance was evaluated by comparing the mean of three biologically independent experiments using a 2-tailed student's *t*-test. Error bars represent ±s.d. A value of P < 0.05 was considered statistically significant. \*\*\*P < 0.001, \*\*P < 0.01, \*P < 0.05. The correlations between TDP-43 expression with the relapse-free survival and overall survival were separately analyzed *via* the online resource Kaplan-Meier Plotter (https://kmplot.com/analysis/index.php?cancer=liver_rnaseq&p=service) and GEPIA (http://gepia.cancer-pku.cn/). In the analysis of HCC tissue microarray, the correlation between TDP-43 and ABHD2 expression was determined according to Pearson's chi-square independent test.

**Reporting summary.** Further information on research design is available in the Nature Research Reporting Summary linked to this article.

## Data availability

The source data underlying the graphs presented in the main figures are shown as Supplementary data 1. Uncropped blots can be found in Supplementary Figs. 7–10. RNA-seq data are available in SRA database of NCBI as PRJNA855537. Plasmids for TDP-43 expression and ABHD3-3′UTR are deposited at Addgene (Accession numbers 190093, 190094, 190095 and 190096). All other data supporting the findings of the study are available within the paper and Supplementary information. Data relating to Supplementary figures are available from Zhi-fa Shen or Bo-wen Liu upon reasonable request.

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

## Acknowledgements
This work was supported by the National Natural Scientific Foundation of China (Grant NO. 81902982), the Science and Technology Key Project of Henan province, China (Grant NO. 212102310179), the Major project of Zhejiang Provincial Natural Science Foundation (Grant NO. LZ22H160007), and the Natural Science Foundation of Henan Province (Grant NO. 222300420264).

## Author contributions
B.L.: Conceptualization, Methodology, Formal analysis, Investigation, Writing-original draft, Visualization, Funding acquisition; X.W. and J.C.: Methodology, Formal analysis, Investigation, Writing-review & editing; L.C., Y.W., B.Z. and J.Z.: Investigation, Validation.; Z.S.: Conceptualization, Supervision, Project administration, Writing-review & editing, Funding acquisition.

## Competing interests
The authors declare no competing interests.
