## [Peer Review File · Communications Biology]

Reviewers' comments:

Reviewer #1 (Remarks to the Author):

The study "TDP-43 upregulates lipid metabolism modulator ABHD2 to suppress apoptosis in hepatocellular carcinoma" from Liu et al found that TDP-43 acts as an RNA-binding protein that regulates the RNA stability of ABHD2 and affects the release of fatty acids and ROS, which in turn regulates apoptosis and affects the growth of liver tumors. The experimental design of this study is reasonable, and the test results can basically support the conclusion. However, some issues need to be clarified, and some additional experimental data need to be added to make the study more complete.

Major.

- ① All Western blot tests need to be analyzed quantitatively, with standard deviation and significance analysis added, and the number of test replicates marked.
- ② All tumor images need to be added with a scale.
- ③ Figure 1: What is the rationale for choosing three cells for different assays, such as interference assays in SMMC-7721 and MHCC97H, but overexpression assays in Huh7?
- ④ Figure 5: The significance analysis of ACTD treatment for 2h in B is **, but the means and standard deviations of the two are close, please confirm this result. In addition, the stability of ACTD changes more in C plot compared with B plot, please confirm this result.
- ⑤ Figure 6, 7: Quantitative analysis should be added to the ROS assay in addition to the fluorescence images, for example, using microplate reader. In addition, a positive control group needs to be added and compared with the treatment group to determine whether the treatment group causes a greater degree of ROS elevation. Since ABHD2 affects apoptosis by influencing ROS levels, a rescue assay, such as adding NAC to remove ROS, is needed to test whether the apoptosis caused by ABHD2 is alleviated. In addition, the expression levels of cellular antioxidant enzymes, such as SOD, CAT, and GPX, should be examined during this process to determine whether TDP-43 and ABHD5 affect the expression of antioxidant enzymes.
- ⑥ Figure 3: The selection of ABHD5 as the study subject lacks sufficient basis. First of all, ABHD5 is not the most decreased gene (Figure 3F), in addition, it is not clear from Figure 3 alone that ABHD5 is a substrate of TDP-43. The heat map presentation of Figure A is not good and cannot clearly see the up- and down-regulation of the gene. B is not significant and can be placed in the Supplementary Information. It is suggested that the authors change the presentation of the results to make the derivation process of ABHD5 clearer.
- ⑦ It is suggested that the authors draw a schematic diagram to summarize the results of this study.
- ⑧ The introduction section of the article needs to be revised well to reflect the innovation and importance of the study. Simply stating that TDP-43 has a role in some cancers but less in hepatocellular carcinoma is very bland and does not interest the reader enough.

Minor

- ① There are errors in some sentences, please check carefully.
- ② Add complete information of reagents. For example ATCC, Wuhan; FBS, Gibco, USA.
- ③ Add genetic background information of mice, etc.
- ④ Add the number of samples and test reproducibility statement, etc.

Reviewer #2 (Remarks to the Author):

The authors demonstrated TDP-43 play important role in HCC. Mechanistically, they showed TDP-43 could bind with the UG-rich sequence¹ of ABHD2 mRNA 3'UTR to enhance the mRNA stability of ABHD2. The story is interesting and most of the experiments are well-designed. However, several issues need to be solved before full acceptance.

1. Why did you choose the two cell lines SMMC-7721 and MHCC97H to knock down TDP-43 and overexpress TDP-43 in Huh-7 cells? Can you provide the expression level of TDP-43 in these cell lines?

2. Page 10, lines 197-200, figure 1H is still using siRNA for knockdown, you just showed stable-knockdown of TDP-43 in figure 1I.
3. A weak overexpression effect can be seen in Figure 4D, but almost no overexpression effect is seen in Figure 1I. Can you provide new evidence to demonstrate the overexpression effect in Figure 1I?
4. The article shows that si-TDP-43#2 has better knockdown effect than si-TDP-43#1, but the protein level in Figure 4D is the opposite, can you explain?
5. What happened in figure 7C, it seems very strange in SMMC-7721 cell line compared to other results you showed, such as figure 1D.
6. In figure 5, one siRNA is not convincing enough, at least for panel B, you should provide results using two siRNAs.
7. Can you show the binding sequences of the two binding sites in figure 5F?
8. Page 13, lines 307, the authors demonstrated that TDP-43 facilitates lipid metabolism via ABHD2, but there seems too little evidence to support it. After TDP-43 knocked down, the FFA content was decreased possibly due to reduced synthesis, reduced intake or other reasons.

Reviewer #3 (Remarks to the Author):

In his work, Liu et al have investigated the role played by TDP-43 in the control of apoptosis in hepatocellular carcinoma (HCC). Specifically, they have focused on the role played by lipid metabolism regulator ABHD2 whose stability is directly regulated by TDP-43. By affecting its expression, TDP-43 can contribute to regulate lipid metabolism in cancer cells and inhibit their apoptosis, thus promoting cancer growth. The role of TDP-43 in cancer progression is a fast growing but still a rather under-studied field. The fact that TDP-43 is involved in neurodegeneration and cancer progression is quite intriguing and nicely underscores the observation that this protein can be involved both in uncontrolled neuronal death but also in uncontrolled cancer progression. In general, therefore, the results of this study will be of interest to researchers of this protein. Nonetheless, several clarifications and additions will be required to make the study more compelling:

- a) as a first observation it is not clear why silencing of TDP-43 was performed in SMMC-7721 and MHCC97H cells whilst overexpression in Huh-7. Has overexpression also been tried for these two previous cell lines?
- b) In Fig.1H-I there seem to be some discrepancies between detected mRNA and protein levels. For example, the mRNA levels of TDP-43 using siRNA#2 are greatly reduced in Fig.1H but the protein does not seem to be greatly diminished in the Western blot (Fig.1I). Moreover, looking at the Western blots for TDP-43 in Fig.1I cells there does not seem to be a lot of overexpression of TDP-43 in the Huh-7 cells. Has this been quantified? For this experiment, authors should provide accurate quantification.
- c) In Fig.2, was silencing in nude mice performed using siRNA#1 or #2? This should be indicated. In addition, why was Ki67 expression investigated in mice and not in vitro (whilst Bcl-2 that was investigated in vitro was not evaluated in mice? To claim that these data are in accordance with the in vitro data (line 217) it would have been good to analyze the same set of genes in the two systems.
- d) Most importantly, patient analysis indicated that the expression level of TDP-43 in HCC tissues was substantially higher than that in adjacent non-tumour tissues (Fig.2I). This is an interesting data, however, as the connection is established with the ABHD2 gene, the authors should report the expression levels of this gene also in the same set of patients. Do the results agree with the author's hypothesis? It would be expected to see that expression levels of this gene in these patients should also be increased in the tumour tissue like TDP-43. Because if they do not this could be a serious problem.
- e) In general, results in Fig.5 are generally convincing in identifying TDP-43 as binding to the Del-1 motif and affecting the stability of this gene. However, if one compares just the WT (black) graphs of Fig.5G and Fig.5H there is an inconsistency in Fig.5G where one would expect the Del-1 fluorescence

to be lower than WT even in the absence of TDP-43 overexpression (as it happens in Fig.5H), for the simple reason that this mutant is not expected to be stabilised by TDP-43 even in normal conditions. Can the authors please clarify?

f) As previously noted, Western blots in Fig.6F and 7F should be quantified as some increased/decreased expressions are not very convincing (ie. Bcl-2 decrease in MHCC97H cell lines).

Reply to Comments

Dear Reviewers:

Thank you very much for your comments for our manuscript (ID COMMSBIO-22-0429), entitled " TDP-43 upregulates lipid metabolism modulator ABHD2 to suppress apoptosis in hepatocellular carcinoma". Your comments are very helpful for revising and improving our paper. We carefully revised our manuscript point-by-point according to the comments and highlighted the changes in the revised manuscript by using colored text. Please find attached reply to comments as follows.

Reviewer #1

Major.

1) **Question:** *All Western blot tests need to be analyzed quantitatively, with standard deviation and significance analysis added, and the number of test replicates marked.*

Answer: Thanks for the suggestion. According to the reviewer's comments, we performed quantification in all Western blot tests and added the standard deviation and significance (Figure S1b-e, S3a, S3b, S4c, S5d, S6b, S6g). The number of test replicates was marked by yellow in the figure legends.

2) **Question:** *All tumor images need to be added with a scale.*

Answer: Thanks for the suggestion. According to the reviewer's comments, we replaced the original tumor images with the same batch of images with a scale.

3) **Question:** *Figure1: What is the rationale for choosing three cells for different assays, such as interference assays in SMMC-7721 and MHCC97H, but overexpression assays in Huh7?*

Answer: We greatly appreciate reviewer's comment. At the beginning of this study, we first detected the expression levels of TDP-43 in different HCC cell lines by qRT-PCR and Western blot assays. The results in these cell lines showed that the expression of TDP-43 was weakest in Huh-7 cell line while abundantly in SMMC-7721 and MHCC97H cell lines (Fig.1).

Accordingly, we chose Huh-7 cells to overexpress TDP-43 and SMMC-7721 and MHCC97H cells to knockdown TDP-43, which is more reasonable and convenient for evaluating the function of TDP-43 in HCC. We added the associated data in the new Supplementary Figure 1 (Figure S1a, b), and revised the Result section marked as yellow in line 183-187.

Fig.1 QRT-PCR (A) and Western blot analysis (B) of the expression levels of TDP-43 in different HCC cell lines.

4) **Question:** *Figure 5: The significance analysis of ACTD treatment for 2h in B is **, but the means and standard deviations of the two are close, please confirm this result. In addition, the stability of ACTD changes more in C plot compared with B plot, please confirm this result.*

Answer: We thank reviewer for indicating this issue. We have checked the raw data and confirmed that there was indeed no statistical significance between the two groups, please forgive our labelling mistake. In order to improve this experiment, we re-designed the experiment by using two siRNAs to knock down TDP-43 and adjusting the treatment dose of siRNA from 25nM to 50nM. The results showed that knockdown of TDP-43 could decrease the mRNA stability of ABHD2 in SMMC-7721 cells (Fig.2). We replaced this data as new Figure 5b in the revised manuscript.

Fig.2 The mRNA stability of ABHD2 was analyzed by qRT-PCR in SMMC-7721 cells upon TDP-43 deletion.

For the second question, the subjects and treatment methods in B plot and C plot are different. The experimental object in B plot is SMMC-7721 cell line while in C plot are MHCC97H-KD-TDP-43 and MHCC97H-Control cell lines. Moreover, in B plot, SMMC-7721 cells were transiently transfected with the indicated siRNAs to silence TDP-43. Based on the above differences, it is normal and acceptable that the decreased degrees of ABHD2 mRNA stability are different in B plot and C plot. We have revised the figure legends of B plot and C plot to describe the experiments in more detail. The changes were marked as yellow in line 624-628.

5) **Question:** *Figure 6, 7: Quantitative analysis should be added to the ROS assay in addition to the fluorescence images, for example, using microplate reader. In addition, a positive control group needs to be added and compared with the treatment group to determine whether the treatment group causes a greater degree of ROS elevation. Since ABHD2 affects apoptosis by influencing ROS levels, a rescue assay, such as adding NAC to remove ROS, is needed to test whether the apoptosis caused by ABHD2 is alleviated. In addition, the expression levels of cellular antioxidant enzymes, such as SOD, CAT, and GPX, should be examined during this process to determine whether TDP-43 and ABHD5 affect the expression of antioxidant enzymes.*

Answer: Thanks for the suggestion. **1.** According to reviewer's comment, we found the positive controls of the same batch of experiments and added them to the corresponding figures (Figure

6c, 7c), and used Image J software to quantify the fluorescence images of ROS (Figure 6b, 7b).

2. According to reviewer's suggestion, we performed the rescue experiment in Huh-7 cells by flow cytometry, qRT-PCR, and Western blot assays. As expectedly, NAC treatment could alleviate the inhibition of apoptosis induced by ABHD2 overexpression (Fig.3), indicating that ABHD2 inhibits apoptosis in an ROS-dependent manner. These results were added to the revised manuscript, please see our revise in Result section (line 306-311) and new Figure S5.

Fig.3 ROS elimination can alleviate the inhibition of apoptosis induced by ABHD2 overexpression.

3. According to reviewer's suggestion, we examined the expression levels of antioxidant enzymes including SOD1, SOD2, CAT, and Gpx in SMMC-7721 cells upon TDP-43 knockdown or ABHD2 knockdown. However, the data showed that silence of TDP-43 or ABHD2 had no impressive effect on the expressions of antioxidant enzymes (Fig.4). This evidence might suggest that TDP-43/ABHD2 affects the content of ROS by promoting its production rather than regulating the scavenging of ROS.

Fig.4 The effect of TDP-43 or ABHD2 on the expressions of antioxidant enzymes.

6) **Question:** *Figure 3: The selection of ABHD5 as the study subject lacks sufficient basis. First of all, ABHD5 is not the most decreased gene (Figure 3F), in addition, it is not clear from Figure 3 alone that ABHD5 is a substrate of TDP-43. The heat map presentation of Figure A is not good and cannot clearly see the up- and down-regulation of the gene. B is not significant and can be placed in the Supplementary Information. It is suggested that the authors change the presentation of the results to make the derivation process of ABHD5 clearer.*

Answer: Thanks for the comments. Sorry for confusing you with the previous presentation in Figure 3. We paid attention to ABHD2 not because it was the most significantly down-regulated gene, but based on the comprehensive judgment of KEGG analysis and qRT-PCR validation results. KEGG pathway enrichment analysis showed that the downregulated genes might be more involved in metabolic pathways (Figure 3d), especially in lipid metabolism pathways (Figure 3e). To further confirm whether TDP-43 was related to metabolism, we screened all metabolism-related genes in down-regulated genes (TOP50) for expression verification. Interestingly, compared with other metabolism-related genes, the lipid metabolism-related gene, abhydrolase domain containing 2 (ABHD2) was the most considerably down-regulated after TDP-43 knockdown (Figure 3f). This evidence was consistent with the analysis of metabolic pathway in Figure 3e. Thus, we concluded that lipid metabolism-related gene ABHD2 was identified as a downstream target of TDP-43. According to reviewer's suggestion, we re-wrote the Result 3 and adjusted the presentation of Figure 3 to make the derivation process of ABHD2 clearer. The changes were highlighted in the revised manuscript by using colored text in line 230-233, 238-244.

7) **Question:** *It is suggested that the authors draw a schematic diagram to summarize the results of this study.*

Answer: Thanks for the comment. According to the reviewer's suggestion, we draw a schematic diagram to summarize the results of our study (Fig.5). As an RNA-binding protein, TDP-43 binds with the 3'UTR of ABHD2 mRNA to enhance its mRNA stability, thereby upregulating ABHD2 expression. Elevated-ABHD2 catalyzes the metabolic process of triglyceride (TG) to free fatty acids (FFAs), thereby promoting β -oxidation pathway and increasing the production of ROS, finally resulting in the suppression of apoptosis and facilitation of proliferation in HCC. The schematic diagram has been added to the revised manuscript in Figure 9. The changes were highlighted in the revised manuscript by using colored text in line 414-419.

Fig.5 Schematic diagram of TDP-43 suppressing apoptosis by modulating ABHD2 in HCC.

8) **Question:** *The introduction section of the article needs to be revised well to reflect the innovation and importance of the study. Simply stating that TDP-43 has a role in some cancers*

but less in hepatocellular carcinoma is very bland and does not interest the reader enough.

Answer: We thank reviewer for the suggestion. We entirely agree with your point. According to your comments, we provided more rationales for the current study in the Introduction section to reflect the importance and innovation of our study. The changes were highlighted by using colored text in the revised manuscript (line 34-38, 55-64).

Minor.

1) **Question:** *There are errors in some sentences, please check carefully.*

Answer: Thanks for your suggestion. According to your comment, we have carefully checked the manuscript and corrected the errors in the sentences.

2) **Question:** *Add complete information of reagents. For example, ATCC, Wuhan; FBS, Gibco, USA.*

Answer: Thank you very much for the suggestion. We carefully checked the contents of the Materials & Methods section and completed the reagent information. The changes were highlighted by using colored text in the revised manuscript (line 97, 121-122).

3) **Question:** *Add genetic background information of mice, etc.*

Answer: Thanks for the suggestion. We added the genetic background information “BALB/c Nude mice, carrying *Foxn1*^{nu} mutation” in Materials & Methods section and marked as yellow in line 140-141.

4) **Question:** *Add the number of samples and test reproducibility statement, etc.*

Answer: Thanks for the suggestion. We added the sentence “All replicates displayed in this paper are biological replicates; technical replicates (usually three) were performed and used to generate the means for each biological replicate. The sample sizes and number of replicates were indicated in the figure legends.” in Materials & Methods section and marked as yellow in line 169-171.

Reviewer #2

1) **Question:** *Why did you choose the two cell lines SMMC-7721 and MHCC97H to knock down TDP-43 and overexpress TDP-43 in Huh-7 cells? Can you provide the expression level of TDP-43 in these cell lines?*

Answer: We thank reviewer for the comment. Please see our responses to the Reviewer#1 (Question 3).

2) **Question:** *Page 10, lines 197-200, figure 1H is still using siRNA for knockdown, you just showed stable-knockdown of TDP-43 in figure 1I.*

Answer: Thanks for the comment. Sorry for the confusion due to our unclear description. In the early stage of the study, we knocked down TDP-43 by transiently transfecting siRNA in SMMC-7721 and MHCC97H cells. Later, we used MHCC97H cells to establish a TDP-43 stable-knockdown cell line (MHCC97H-KD-TDP-43) and its control cell line (MHCC97H-Control), and applied them in the subsequent studies. The cell line used in Figure 1h is SMMC-7721; so, the siRNAs were used for TDP-43 knockdown. The “MHCC97H” cell line used in Figure 1i is actually the above-mentioned MHCC97H-KD-TDP-43 and MHCC97H-Control cell lines. In order to unify the labeling form, it was therefore expressed as “Control and KD-TDP-43”. We modified the description in the Result section and figure legends to make the content clearer. The changes were marked as yellow in line 196-200, 564-569.

3) **Question:** *A weak overexpression effect can be seen in Figure 4D, but almost no overexpression effect is seen in Figure 1I. Can you provide new evidence to demonstrate the overexpression effect in Figure 1I?*

Answer: Thanks for the comment. We performed quantification analysis in Figure 1i and found that the relative intensity of blot in overexpression group was higher than that in vector group ($p < 0.01$) (Fig.6A). This evidence indicated the overexpression effect in Figure 1i. Since the same batch of protein samples in Figure 1i has been used up, in order to demonstrate the overexpression efficiency, we performed transfection and Western blot analysis again in Huh-7 cells. The result showed an effective overexpression of TDP-43 in Huh-7 cells (Fig.6B).

Fig.6 The confirmation of overexpression effect of TDP-43 in Huh-7 cells.

4) **Question:** *The article shows that si-TDP-43#2 has better knockdown effect than si-TDP-43#1, but the protein level in Figure 4D is the opposite, can you explain?*

Answer: Thank you very much for your comment. As you said, the si-TDP-43#2 had better knockdown effect than si-TDP-43#1, which was obviously demonstrated by qRT-PCR assay. But the differences in protein levels seemed less pronounced. **1.** To clarify this issue, we performed quantification analysis in Figure 1i and 4d. In Figure 1i, the interference effect of si-TDP-43#2 was slightly stronger than that of si-TDP-43#1, but there was no statistical difference between them (Fig.7). In Figure 4d, the knockdown effect of si-TDP-43#2 was weaker than that of si-TDP-43#1, however, there was still no statistical difference between them (Fig.7).

Fig.7. Quantification analysis of TDP-43 in Figure 1i and Figure 4d.

2. To further verify the knockdown effect of siRNAs, we performed three independent experiments again by transiently transfecting siRNAs or si-Control into SMMC-7721 cells, and

detected the interference efficiency by qRT-PCR and Western blot assays. The results showed that si-TDP-43#2 still displayed a stronger interference effect at the mRNA level, but there was no significant difference between them at the protein level (Fig.8).

Fig.8 Confirmation of the interference efficiency of si-TDP-43s by qRT-PCR and Western blot assays.

3. In response to this phenomenon, we speculate that it may be related to the regulatory mechanism at the post-transcriptional or translational level, which requires more in-depth research to elucidate. In this study, we focused on the role of TDP-43 in HCC. We used two different siRNAs to successfully silence TDP-43 and objectively presented the results to reveal the function and mechanism of TDP-43 in the malignant progression of HCC.

5) **Question:** What happened in figure 7C, it seems very strange in SMMC-7721 cell line compared to other results you showed, such as figure 1D.

Answer: Thank you for pointing out this important issue. Sorry, we didn't notice the anomaly

in this data earlier. According to your comments, we reviewed the experimental records at that time and found that the cell state was not very good at that time, which may cause a large number of cell necrosis after transfection. To solve this problem, we re-prepared SMMC-7721 cells, transfected them with the same dose as before and detected them by flow cytometry. The new result indicated that overexpression of ABHD2 markedly alleviated the apoptosis caused by TDP-43 knockdown (Fig.9). In the revised manuscript, the previous Figure 7c will be replaced by this new data.

Fig.9 Flow cytometry analysis of apoptosis in SMMC-7721 cells after transient transfection with the displayed siRNAs and plasmids and the statistical graphs.

6) **Question:** In figure 5, one siRNA is not convincing enough, at least for panel B, you should provide results using two siRNAs.

Answer: Thank you very much for the suggestion. We re-performed the experiment in Figure 5b by using two siRNAs to silence TDP-43. Please refer to Fig.2 in this reply letter. We have replaced this data as new Figure 5b in the revised manuscript.

7) **Question:** Can you show the binding sequences of the two binding sites in figure 5F?

Answer: Thanks for the comment. According to your suggestion, we showed the binding sequences of the two binding sites in Figure 5f and supplemented the specific location information of binding sequences in figure legend (line 633-634).

8) **Question:** *Page 13, lines 307, the authors demonstrated that TDP-43 facilitates lipid metabolism via ABHD2, but there seems too little evidence to support it. After TDP-43 knocked down, the FFA content was decreased possibly due to reduced synthesis, reduced intake or other reasons.*

Answer: We greatly appreciate the reviewer's comment which is very helpful for improving our article. To further demonstrate our conclusion, we performed the following experiments: **1.** ABHD2 can participate in lipid metabolism by exerting triacylglycerol lipase activity; under the action of triacylglycerol lipase, triacylglycerols are hydrolyzed to produce fatty acids. Therefore, we examined the effects of ABHD2 and TDP-43 on intracellular triglyceride (TG) levels. The data displayed that deletion of ABHD2 obviously induced the accumulation of TG in SMMC-7721 and MHCC97H cells (Fig.10A, B). Notably, TDP-43 knockdown also induced the accumulation of TG while overexpression of ABHD2 could alleviate this phenomenon (Fig.10C, D). **2.** Considering that the anti-apoptosis effect of TDP-43 was mediated by ABHD2-related lipid metabolism, next, we intended to evaluate the role of metabolites FFA in TDP-43-suppressed apoptosis. The SMMC-7721 cells were transiently transfected with si-Control or si-TDP-43#2 along with the treatment of DMSO or FFA (oleic acid, 30 μ M). The result of flow cytometry, qRT-PCR, and Western blot assays showed that FFA treatment efficiently reverse the apoptosis caused by TDP-43 deletion (Fig.10E-H). These evidences further confirmed the vital role of ABHD2-related lipid metabolism in TDP-43-suppressed apoptosis.

Fig.10. TDP-43 promotes lipid metabolism via ABHD2 to suppress apoptosis.

3. As you said, the reduction of FFA upon TDP-43 knockdown is also possibly due to reduced synthesis or intake. We totally agree with your point. To address this issue, we initially selected several key molecules from these pathways for detection, including fatty acid synthesis (ACC1, FASN, SCD) and fatty acid uptake (CD36). QRT-PCR assay showed that TDP-43 had no obvious effect on the expression of these genes (Fig.11). Despite this preliminary data, whether TDP-43 can affect these pathways requires more in-depth study. In our study, we screened out ABHD2, a key enzyme in lipolysis, by RNA-seq results. Therefore, we mainly focused on the lipolytic aspect. We found that TDP-43 was able to promote the breakdown of TG through

ABHD2, leading to the upregulation of FFA. These evidences directly demonstrated that TDP-43 promoted lipid metabolism *via* ABHD2. In the revised manuscript, Fig.10 was added as new Figure S4a, S4b, S6a, and S6d-g; meanwhile, we supplemented the associated content in the Result section marked as yellow in line 290-294, 315-319, 327-333, 381-382, 388-389.

Fig.11 QRT-PCR assay of the expression level of lipid metabolism-related genes in SMMC-7721 cells after transient transfection with the displayed siRNAs.

Reviewer #3

1) **Question:** *as a first observation it is not clear why silencing of TDP-43 was performed in SMMC-7721 and MHCC97H cells whilst overexpression in Huh-7. Has overexpression also been tried for these two previous cell lines?*

Answer: Thanks for the comment. We apologize for lack of clarity about this issue. Please see our responses to the Reviewer#1 (Question 3).

2) **Question:** *In Fig.1H-I there seem to be some discrepancies between detected mRNA and protein levels. For example, the mRNA levels of TDP-43 using siRNA#2 are greatly reduced in Fig.1H but the protein does not seem to be greatly diminished in the Western blot (Fig.1I). Moreover, looking at the Western blots for TDP-43 in Fig.1I cells there does not seem to be a lot of overexpression of TDP-43 in the Huh-7 cells. Has this been quantified? For this experiment, authors should provide accurate quantification.*

Answer: We thank reviewer for the comment. As you suggested, we performed quantification analysis for Western blots in Figure 1i and added the results in the revised manuscript (Figure S1c-e). For details, please see our responses to the Reviewer#2 (Question 3 and 4).

3) **Question:** *In Fig.2, was silencing in nude mice performed using siRNA#1 or #2? This should be indicated. In addition, why was Ki67 expression investigated in mice and not in vitro (whilst Bcl-2 that was investigated in vitro was not evaluated in mice? To claim that these data are in accordance with the in vitro data (line 217) it would have been good to analyze the same set of genes in the two systems.*

Answer: Thanks for the comment. In Figure 2, sh-TDP-43#2 was used for silencing TDP-43 in nude mice. The sequence of sh-TDP-43#2 was presented in Supplementary table and the detailed description was added to Supplementary materials and methods. The change was marked as yellow. For the second question, we really appreciate your suggestion. In order to increase the in vitro analytical evidence of Ki67, we supplemented Western blot analysis of Ki67 in Figure.1i, S1c-e, 7g, and S6b. Meanwhile, we also performed IHC staining of Bcl-2 by separately using the same batch of samples in Figure 2e and 8e. In the revised manuscript, the above results of Bcl-2 were supplemented in Figure.2e, 2f, 8e, and 8f.

4) **Question:** *Most importantly, patient analysis indicated that the expression level of TDP-43 in HCC tissues was substantially higher than that in adjacent non-tumour tissues (Fig.2I). This is an interesting data, however, as the connection is established with the ABHD2 gene, the authors should report the expression levels of this gene also in the same set of patients. Do the results agree with the author's hypothesis? It would be expected to see that expression levels of this gene in these patients should also be increased in the tumour tissue like TDP-43. Because if they do not this could be a serious problem.*

Answer: Thank you for your comments. Your suggestion is very valuable for our manuscript. We can understand your concern. During our research, we also thought about the same issue and analyzed the expression level of ABHD2 in GSE36376. The results are shown in the Fig.12.

Fig.12 Analysis of ABHD2 expression in HCC tumors and adjacent non-tumor tissues in GSE36376.

We analyzed all three probes targeting ABHD2 in GSE36376, and the results showed that the expression level of ABHD2 in adjacent tissues was slightly higher than that in HCC tissues (Fig.12). In view of this phenomenon, we have conducted in-depth thinking and research, and believe that this result is reasonable. Now we mainly explain it from the following three points:

1. According to the gene correlation analysis in online database GEPIA (<http://gepia.cancer-pku.cn/detail.php>), we found that the expression of TDP-43 and ABHD2 in HCC was highly correlated (Fig.13). This evidence preliminarily confirmed our conclusion that TDP-43 and ABHD2 were closely related in HCC.

Fig.13 Expression correlation analysis of TDP-43 (TARDBP) and ABHD2 in HCC

2. When considering the problem of gene expression in tissue samples, more and more studies at the single cell level suggest that we need to pay attention to tissue heterogeneity [1]. RNA-seq of tissue samples (e.g. GSE36376) represents the average difference of total RNA across all cell types, and cannot directly represent the RNA difference in liver cancer

parenchymal cells [1]. In tumor tissue, there is an immune microenvironment with abnormal infiltration of immune cells, which is difficult to be ignored [2]. We used the proteinatlas database (<https://www.proteinatlas.org/humanproteome/single+cell+type>) to analyze the expression of TDP-43 and ABHD2 at the single cell level in liver. Interestingly, TDP-43 was abundantly expressed in all types of immune cells and hepatocytes; while ABHD2 was only abundantly expressed in hepatocytes, and was weakly expressed in immune cells (Fig.14). So, the expression of the two genes in tissues will be affected by the level of immune cell infiltration. ABHD2 is not up-regulated in tissue samples, which does not directly mean that it is not up-regulated in liver cancer parenchyma cells. Therefore, it is biased to evaluate the role of ABHD2 in HCC solely from the level of total tissue RNA.

Fig.14 Analysis of the expression of TDP-43 and ABHD2 at the single cell level in liver.

3. Our research is based on RNA-seq implemented by liver parenchymal tumor cells. This research method can eliminate the interference of genes carried by immune cells in the measurement of tissue sample expression. Through RNA-seq and expression regulation experiments of liver cancer parenchymal cells, we gradually found that ABHD2 is the target gene of TDP-43, and further clarified the molecular mechanism of TDP-43 regulating ABHD2 in HCC cells. Moreover, multiple in vivo and in vitro data also suggest that ABHD2 has the function of promoting the malignant phenotype of HCC. These results can directly reveal the regulatory relationship between TDP-43 and ABHD2 in HCC.

To sum up, we believe that although the expression level of ABHD2 does not show significant differences in the tissue samples of GSE36376, it does not seem to affect the hypothesis that ABHD2 plays a role in the progression of HCC. Finally, we supplemented the results of gene expression correlation (Fig.13) in the Figure 4 in the revised manuscript. The changes in Result section were marked as yellow in line 252-254, 612-614.

5) Question: *In general, results in Fig.5 are generally convincing in identifying TDP-43 as binding to the Del-1 motif and affecting the stability of this gene. However, if one compares just the WT (black) graphs of Fig.5G and Fig.5H there is an inconsistency in Fig.5G where one would expect the Del-1 fluorescence to be lower than WT even in the absence of TDP-43 overexpression (as it happens in Fig.5H), for the simple reason that this mutant is not expected to be stabilised by TDP-43 even in normal conditions. Can the authors please clarify?*

Answer: We thank reviewer for the comment. Your question is very good, and we also find this phenomenon interesting. Therefore, we have carried out the following research on this problem:

1. We first examined the raw data in Figure 5g and 5h and found that both results showed that the activity of Del-1 was lower than that of WT. However, due to the difference in tick interval of Y-axis in the two figures, the difference between WT and Del-1 in Figure 5g is not easy to observe. Notably, in Figure 5g, there was no significant difference ($p=0.1028$) between WT group (mean: 1) and Del-1 group (mean: 0.8923). In Fig.5h, the statistical significance between WT group (mean: 1) and Del-1 group (mean: 0.5503) was ** ($p=0.0041$). The original data was presented in Supplementary data 1.

2. Then, we performed luciferase reporter gene assay in HEK 293T cells to directly detect the activity difference among WT, Del-1 and Del-2. The results showed that the activity of Del-1 (mean: 0.8142) was indeed lower than that of WT (mean:1) without any treatment, but the difference was not statistically significant ($p=0.0808$) (Fig.15A).

3. In view of the above results, we repeated the experiments of Figure 5g and 5h (Fig.15B, C). In Fig.15B, there was still no significant difference ($p=0.0817$) between WT group (mean: 1) and Del-1 group (mean: 0.8619). In Fig.15C, the statistical significance between WT group (mean: 1) and Del-1 group (mean: 0.6929) was * ($p=0.0114$). This result was consistent with Figure 5g and 5h in our original manuscript.

Fig.15 Luciferase reporter gene assay of ABHD2 3'UTR activity in HEK 293T cells.

Taking the above results together, as you said, the activity of Del-1 is lower than that of WT. However, the magnitude of the difference between them was different in distinct treatments. Although HEK 293T cells were used, in Figure 5g, the control group (black column) was transfected with empty vector (pEGFP-C1), while in Figure 5h, si-Control was transfected. Therefore, we speculated that the difference in transfection system might cause the difference between WT and Del-1 to fluctuate. But overall, these results consistently show that the activity of Del-1 is lower than that of WT, and the existing data are sufficient to support our conclusion that “TDP-43 regulates the stability of ABHD2 mRNA through UG-rich sequence 1”.

6) **Question:** As previously noted, Western blots in Fig.6F and 7F should be quantified as some increased/decreased expressions are not very convincing (ie. Bcl-2 decrease in MHCC97H cell lines).

Answer: Thanks for the comment. According to your suggestion, we performed quantification

analysis for Western blots in Figure 6g and 7g (Figure 6f and 7f in the original version) and added the results in the revised manuscript (Figure S4c and S6b).

References

- [1] A.L. Chu, J.D. Schilling, K.R. King, A.E. Feldstein, The Power of Single-Cell Analysis for the Study of Liver Pathobiology, *Hepatology*, 73 (2021) 437-448.
- [2] Y. Kurebayashi, H. Ojima, H. Tsujikawa, N. Kubota, J. Maehara, Y. Abe, M. Kitago, M. Shinoda, Y. Kitagawa, M. Sakamoto, Landscape of immune microenvironment in hepatocellular carcinoma and its additional impact on histological and molecular classification, *Hepatology*, 68 (2018) 1025-1041.

REVIEWERS' COMMENTS:

Reviewer #1 (Remarks to the Author):

I see that the authors have made serious revisions according to the reviewers' comments. The quality of the whole article has been greatly improved. Many thanks to the authors and the team members for their efforts on this article. I think this study is now ready for publication.

Reviewer #2 (Remarks to the Author):

Authors have added additional experiments that have made their manuscript more convincing.

Reviewer #3 (Remarks to the Author):

Authors have answered well to all queries from this reviewer and the manuscript has been considerably improved for internal consistency of results.